# A Smart ANN-Based Converter for Efficient Bidirectional Power Flow in Hybrid Electric Vehicles

R.S.Ravi Sankar [1], Keerthi Deepika.K [1,*], Mohammad Alsharef [2] and Basem Alamri [2]

1 Department of Electrical & Electronics Engineering, Vignan's Institute of Information Technology, Visakhapatnam 530046, India
2 Department of Electrical Engineering, College of Engineering, Taif University, P.O. Box 11099, Taif 21944, Saudi Arabia
* Correspondence: keerthi.deepika@gmail.com or kkdeepika@vignaniit.edu.in

**Abstract:** Electric vehicles (EV) are promising alternate fuel technologies to curtail vehicular emissions. A modeling framework in a hybrid electric vehicle system with a joint analysis of EV in powering and regenerative braking mode is introduced. Bidirectional DC–DC converters (BDC) are important for widespread voltage matching and effective for recovery of feedback energy. BDC connects the first voltage source (FVS) and second voltage source (SVS), and a DC-bus voltage at various levels is implemented. The main objectives of this work are coordinated control of the DC energy sources of various voltage levels, independent power flow between both the energy sources, and regulation of current flow from the DC-bus to the voltage sources. Optimization of the feedback control in the converter circuit of HEV is designed using an artificial neural network (ANN). Applicability of the EV in bidirectional power flow management is demonstrated. Furthermore, the dual-source low-voltage buck/boost mode enables independent power flow management between the two sources—FVS and SVS. In both modes of operation of the converter, drive performance with an ANN is compared with a conventional proportional–integral control. Simulations executed in MATLAB/Simulink demonstrate low steady-state error, peak overshoot, and settling time with the ANN controller.

**Keywords:** hybrid electric vehicle (HEV); two battery sources; bidirectional DC/DC converter; artificial neural network; internal combustion engine (ICE)

## 1. Introduction

Nowadays, transportation systems play a crucial role in the entire world. The majority are automobiles with internal combustion engines (ICE). Using ICEs has resulted in acute issues including air pollution, global warming, and rapid depletion of the world's petroleum resources [1]. The three types of vehicles suggested to replace conventional cars with ICE are fuel cell vehicles (FCV) [2], electric vehicles (EV) [3] and hybrid electric vehicles (HEV). The performance of fuel cell and electric vehicles falls well short of what is required. As a result, the focus of advanced vehicle technology development has shifted to HEVs [4]. A hybrid vehicle has two or more forms of energy stored on board: one is a specific type of gasoline is used as fuel in a conventional hybrid electric car. The other is an electrical storage device that can be used in both directions. There are various methods to minimize fuel consumption in hybrid electric vehicles [5]. By utilizing an energy storage system, hybrid electric vehicles can reduce fuel consumption in a variety of ways, including collecting energy during braking, downsizing the engine, operating the engine more effectively, and turning off the engine when it is not in use [6]. Regenerative braking is used in advanced HEVs to convert the vehicle's kinetic energy into electric energy rather than dissipating it as heat energy as per standard brakes [7]. A few HEVs [8] produce energy by spinning an electrical generator (also called a motor–generator), which is then used to charge their batteries or directly power the electric drive motors. A hybrid electric car produces fewer emissions from its ICE than a gasoline car of comparable size, further improving fuel

economy. Moreover, it contains a component with a high energy density, such as a super capacitor (SC), which avoids peak energy transience during acceleration and regenerative braking systems [9]. SCs may store and release regenerative energy during deceleration and acceleration, producing extra power. Figure 1 shows a basic block diagram for a hybrid electric vehicle (HEV) power system.

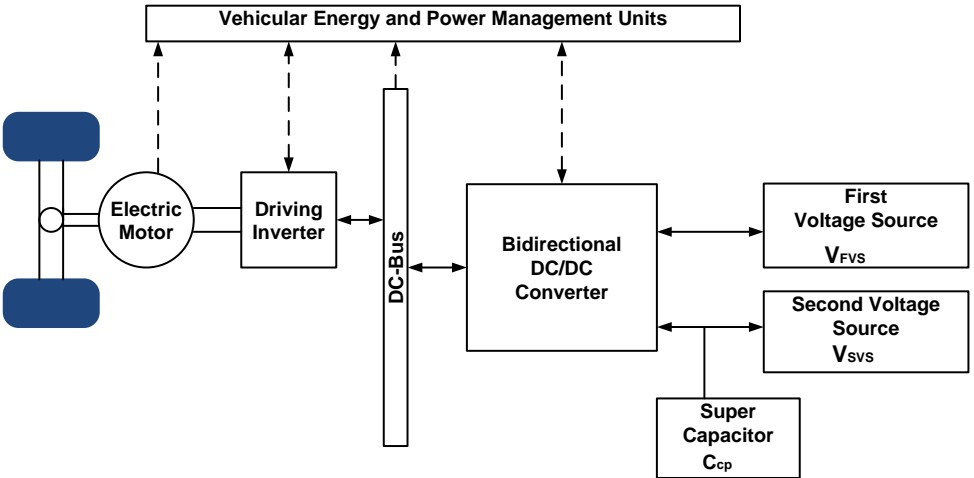

**Figure 1.** Basic block diagram for a hybrid electric vehicle (HEV) power system.

Power conversion efficiency is required by modern electronic equipment that operates at lower voltages and higher currents. Power converters are a cost-effective way to obtain a regulated voltage from a standard power source. Grid-connected power converters are popular for providing uninterrupted power and improved power quality [10]. A DC–DC converter using two back-to-back boost converters for bidirectional power flow in DC and hybrid microgrids are presented in [11]. In [12] Yongseok Choi et al. (2007) looked at the problem of energy conservation while taking into account the DC–DC converter's power consumption characteristics.

The main objective of the DC–DC converter is to adjust the output of the front-end AC–DC converter and to charge the EV in the desired mode (CC or CV). The most common DC–DC converter topologies include voltage-fed bridges; current-fed bridges; appropriate combinations of these; and resonant converters [13,14]. The number of active switches, and thereby device stress, is reduced in dual active voltage-fed full bridges in comparison to voltage- and current-fed full-bridges. Unidirectional DC–DC converters were studied in [15] for power decoupling between the fuel cell and DC-bus.

Considering the diverse vehicle driving settings, widespread voltage matching and decoupling of real and reactive power are crucial. These can be effectively ensured with the bidirectional DC–DC converter. Drawbacks in the operation of conventional BDCs are reported in the literature [16,17] from different viewpoints, although this paper does not aim to address them. Previous studies proposed several isolated and non-isolated bidirectional topologies to improve dynamic performance, gain, efficiency, and operability of BDCs for energy storage and renewable applications. In [18], a bidirectional DC–DC converter for power decoupling in a distribution system with a PV system and electric springs was investigated. A bidirectional DC–DC converter used to regulate the charging current in a bidirectional EV charger is discussed in [19,20]. These operate with isolated and non-isolated circuit arrangements. Another major benefit is reduction in volume, weight, and cost of the charger. In this context, several topologies of bidirectional converters specifically applied to electric vehicles have been investigated and reviewed [21]. A two-phase interleaved bidirectional DC/DC converter was studied in [22]. A circuit configuration with the aim of an increased voltage conversion ratio was proposed in [23]. In [24], a dual active bridge bidirectional converter for enhanced power range for an ultracapacitor was designed. A multi-port concept for a bidirectional power converter with a battery/supercapacitor

was extensively simulated in [25]. A bidirectional DC/DC converter with dual-battery energy storage for a hybrid electric vehicle system was developed in [26].

The main objectives of this work are coordinated control of the DC energy sources of various voltage levels, independent power flow between both the energy sources, and regulation of current flow from the DC-bus to the voltage sources. This work aims to optimise the converter control to investigate under diverse combinations of the voltage levels of the sources, energy flow between the sources, modes of operation, and inductor currents.

Research by Kang et al. [27] on the charging system of a hybrid EV (HEV) detailed the PI control methods for DC–DC converters to improve stability. A neural network controller was employed to control the interleaved boost DC–DC converter associated with a proton exchange membrane fuel cell in [28]. In [29], a neural network was implemented in the energy management system in electric vehicles using ultra capacitors. Fuzzy neural network PID control was developed in [30] in the pressure control of the EVs. In [31], Wang et al. proposed a method using back propagation neural networks in estimation of the state of health of the battery in electric vehicles. Reddy and Sudhakar et.al. [32] designed an ANFIS-based controller in the EV drive train to extract maximum power from the fuel cells. Wang et. al. [33] applied an adaptive sliding-mode control for current control in the boost converters in EVs. Liu et al. [34] analysed the lithium-ion battery dynamics in a multi-objective function framework.

In this paper, the objectives of the proposed bidirectional DC–DC converter and its controller are regulation of the energy flow between voltage sources as well as the mitigation of the ripples in inductor currents. The proposed solution deploys a neural network controller to generate the required duty cycle. The advantage of a neural network controller is its simplicity, with a limited number of tests required to construct it. Proposed PWM allows minimizing the ripple of current for all voltage levels. This paper is organized as follows: In Section 2, the architecture and operating modes of the dual converter are elaborated on. The control technique of the converter is detailed in Section 3. In Section 4, the validation of the proposed vehicle is projected with simulation results. Finally, a comparison of PI and an ANN is presented in Section 5.

## 2. Architecture and Operating Modes

The illustration, $V_{HB}, V_{FVS}, V_{SVS}$ represents high-bus voltage, first voltage source (*FVS*), and second voltage source (SVS), respectively. The FVS and SVS control loops were turned on and off using two bidirectional power switches in this architecture. Voltage gain between low-voltage sources $V_{FVS}$ *and* $V_{SVS}$ is developed using a pump capacitor ($C_{cp}$) that separates the voltage with active switches (M$_1$, M$_2$, M$_3$, M$_4$) and two inductors (L$_f$, L$_s$). $C_{cp}$ reduces voltage stress across switches and thereby eliminates the need for an extremely high duty ratio. As indicated in Figure 2, bidirectional switches ($S_w, S_{FVS}, S_{SVS}$) are MOS-FETS connected in obverse direction. They enable the circuit's four-quadrant functioning, which allows the flow of power control between two low-voltage sources- $V_{FVS}$ *and* $V_{SVS}$. Additionally, they are responsible for the suppression of positive and negative voltages. Table 1 describes the bidirectional concepts and operation modes of the circuit.

**Table 1.** Conduction status of devices for various operation modes.

| Operation Modes | Switches in ON | Switches in OFF | Control Switches | Synchronous Rectifiers (SR) |
|---|---|---|---|---|
| Dual low-voltage sources powering mode (Accelerating, $x_1 = 1$, $x_2 = 1$) | S$_{FVS}$,S$_{SVS}$ | S$_w$ | M$_3$,M$_4$ | M$_1$,M$_2$ |
| DC-bus energy-regenerative mode at high-voltage (Braking, $x_1 = 1$, $x_2 = 1$) | S$_{FVS}$,S$_{SVS}$ | S$_w$ | M$_1$,M$_2$ | M$_3$,M$_4$ |
| Dual low-voltage sources buck mode (FVS to SVS, $x_1 = 0$, $x_2 = 0$) | S$_{FVS}$,S$_{SVS}$ | M$_1$,M$_2$,M$_4$ | S$_w$ | M$_3$ |
| Dual low-voltage sources boost mode (FVS to SVS, $x_1 = 0$, $x_2 = 0$) | S$_{FVS}$,S$_{SVS}$ | M$_1$,M$_2$,M$_4$ | M$_3$ | S$_w$ |

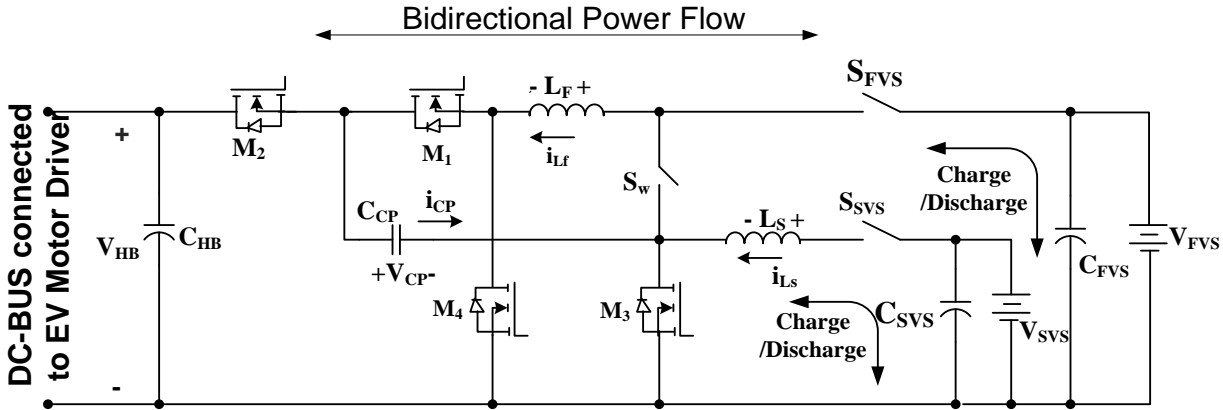

**Figure 2.** Dual-battery energy storage in BDC architecture.

*2.1. Dual Low-Voltage Sources Powering Mode*

Switches ($S_{FVS}$ and $S_{SVS}$) are turned ON. Dual low voltage sources ($V_{FVS}$, $V_{SVS}$) provide energy to connected loads and the DC-bus, as shown in Figure 3a. $M_3$ and $M_4$ switches on the bottom side are controlled by active switching with a 180° phase shift angular position, whereas $M_1$ and $M_2$ switches on the top side are synchronous rectifiers.

Four circuit stages are featured in Figure 4. Dual-sources low voltage powering mode steady-state waveforms with a duty ratio greater than 50%, as illustrated in Figure 3b, and the operation, can be stated briefly as follows.

(a) Stage 1 [$t_0 < t < t_1$]: Time period at this point is $(1 − D_t)T_s$. As shown in Figure 4a, switches $M_1$ and $M_3$ are ON, whereas switches $M_2$ and $M_4$ are OFF. Voltage across the first inductor $L_f$, which drops linearly from its original value, is represented by the differential between the charge pump voltage, $V_{CP}$ and the lower side voltage , $V_{FVS}$. Voltage across the second inductor $L_s$ charged by the energy source, $V_{SVS}$ increases linearly. Characteristic equations in stage 1 in terms of voltage across the inductors $L_f$ and $L_s$ are denoted by Equations (1) and (2):

$$L_f \frac{di_{Lf}}{dt} = V_{SVS} − V_{CP} \tag{1}$$

$$L_s \frac{di_{Ls}}{dt} = V_{SVS} \tag{2}$$

(b) Stage 2 [$t_1 < t < t_2$]: Time interval during this stage is $(D_t − 0.5)T_s$. $M_3$ and $M_4$ switches are ON, while $M_1$ and $M_2$ switches are OFF as shown in Figure 4b. Lower side voltages, $V_{FVS}$, $V_{SVS}$, are located between the first and second inductors. Inductor currents increase linearly. Characteristic equations in stage 2 in terms of voltage across the inductors $L_f$ and $L_s$ are denoted by Equations (3) and (4):

$$L_f \frac{di_{Lf}}{dt} = V_{FVS} \tag{3}$$

$$L_s \frac{di_{Ls}}{dt} = V_{SVS} \tag{4}$$

(c) Stage 3 [$t_2 < t < t_3$]: The time interval at this stage is $(1 − D_t)Ts$. $M_1$ and $M_3$ switches are off, while $M_2$ and $M_4$ switches are on, as shown in Figure 4c. Characteristic equations in stage 3, in terms of voltage across the inductors $L_f$ and $L_s$, are denoted by Equations (5) and (6):

$$L_f \frac{di_{Lf}}{dt} = V_{FVS} \tag{5}$$

$$L_s \frac{di_{Ls}}{dt} = V_{CP} + V_{SVS} − V_{HB} \tag{6}$$

(d) Stage 4 [$t_3 < t < t_4$]: Time interval at this stage is $(D_t - 0.5)T_s$. $M_3$ and $M_4$ switches are on, while the $M_1$ and $M_2$ switches are off, as shown in Figure 4d. Characteristic equations in stage 4, in terms of voltage across the inductors $L_f$ and $L_s$, are denoted by Equations (7) and (8):

$$L_f \frac{di_{Lf}}{dt} = V_{FVS} \tag{7}$$

$$L_s \frac{di_{Ls}}{dt} = V_{SVS} \tag{8}$$

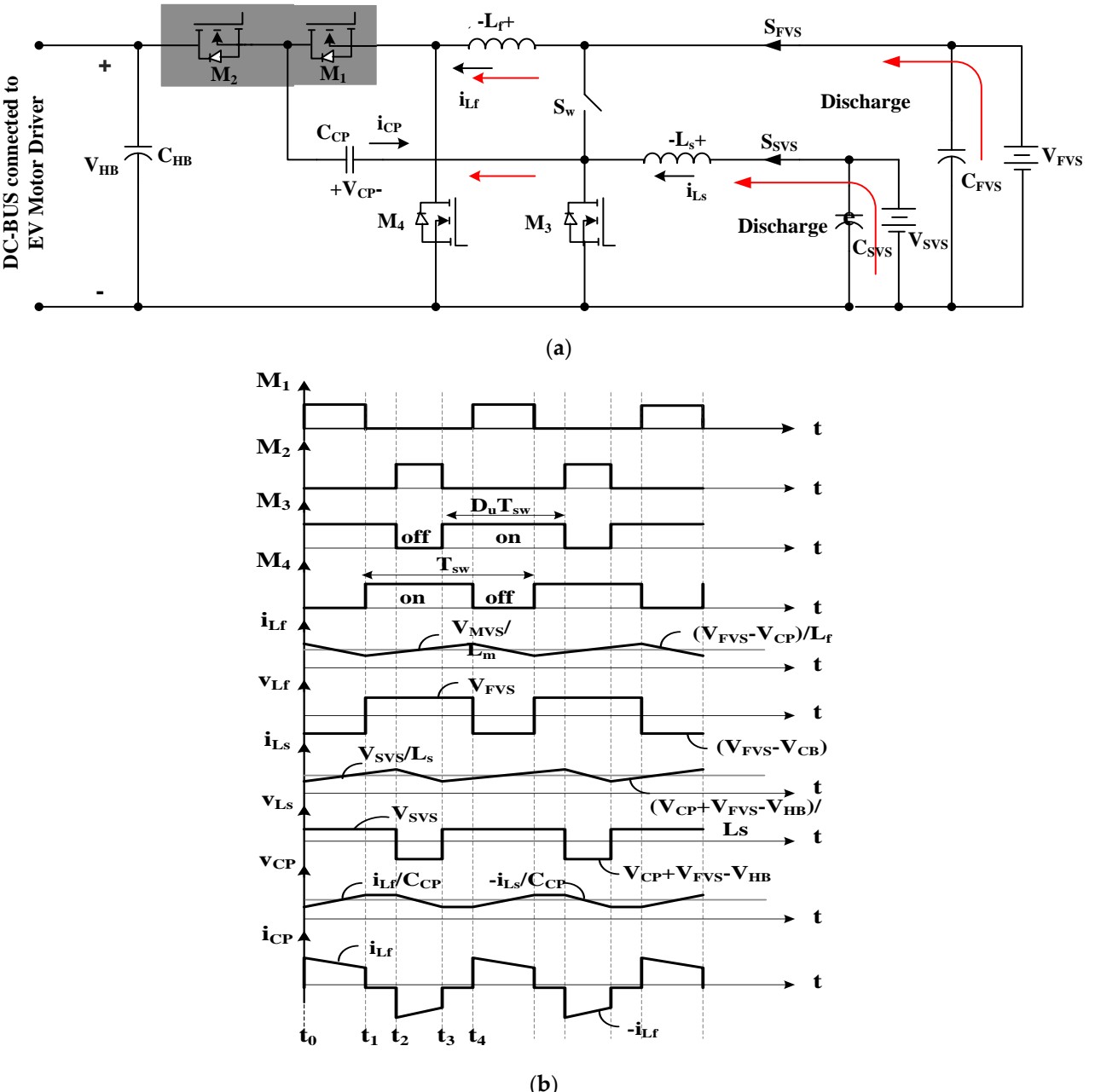

(**a**)

(**b**)

**Figure 3.** (**a**) Schematic Diagram. (**b**) Waveforms of BDC's dual low-voltage sources powering mode.

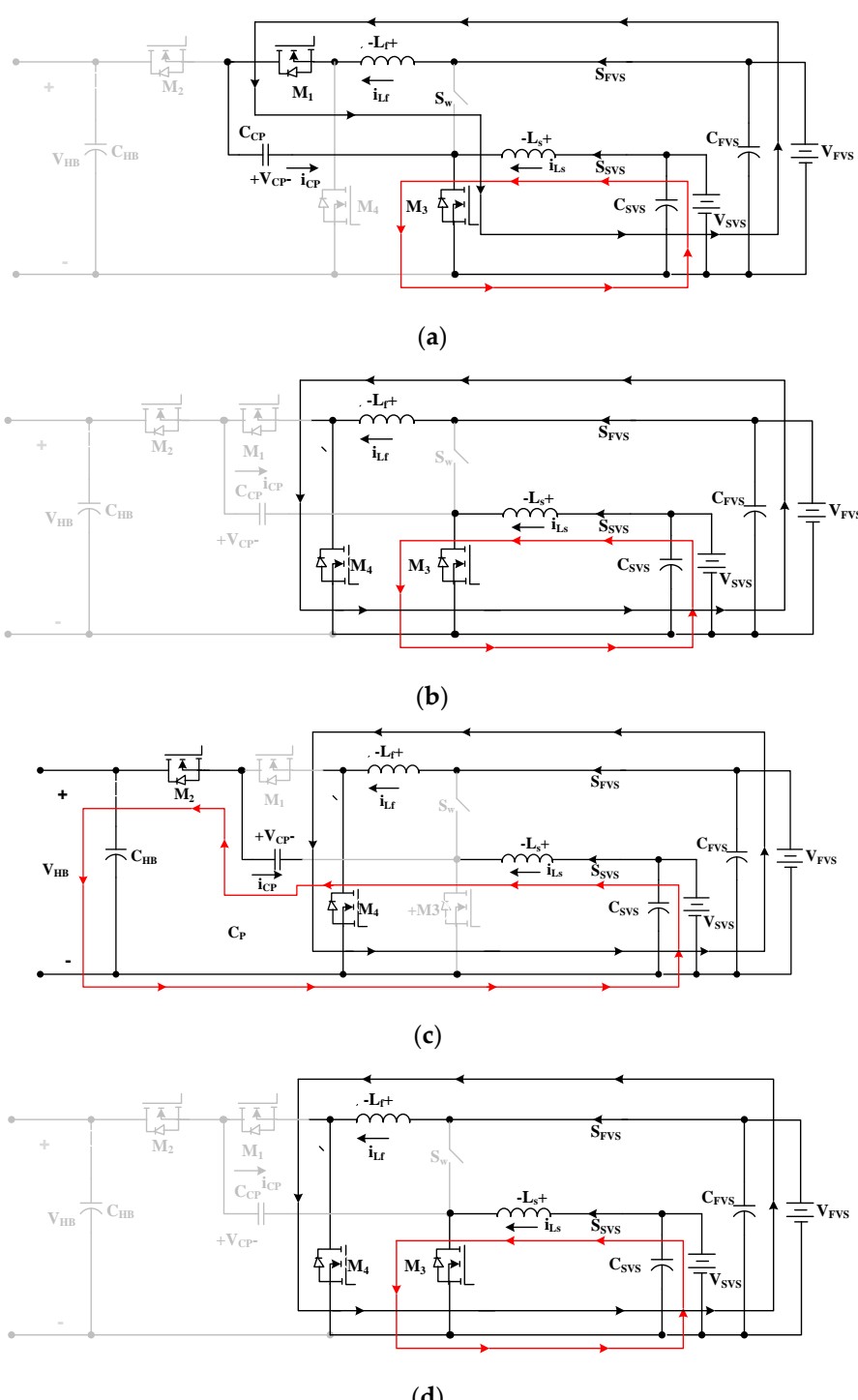

**Figure 4.** Circuit stages of BDC's dual low-voltage sources powering mode.

## 2.2. Regenerative Mode at High Voltage

In regenerative mode, the motor's kinetic energy is returned to the voltage sources. Regeneration energy can be more than the battery's storage capability, wherein the additional energy will be used to charge the storage devices. On the higher side, active switching regulates the current through the inductors, i.e., $M_3$ and $M_4$ switches with 180° phase shift angle. However, the lower side switches- $M_1$ and $M_2$, act as synchronous rectifiers, increasing the converter's conversion performance. This can be observed in Figure 5a.

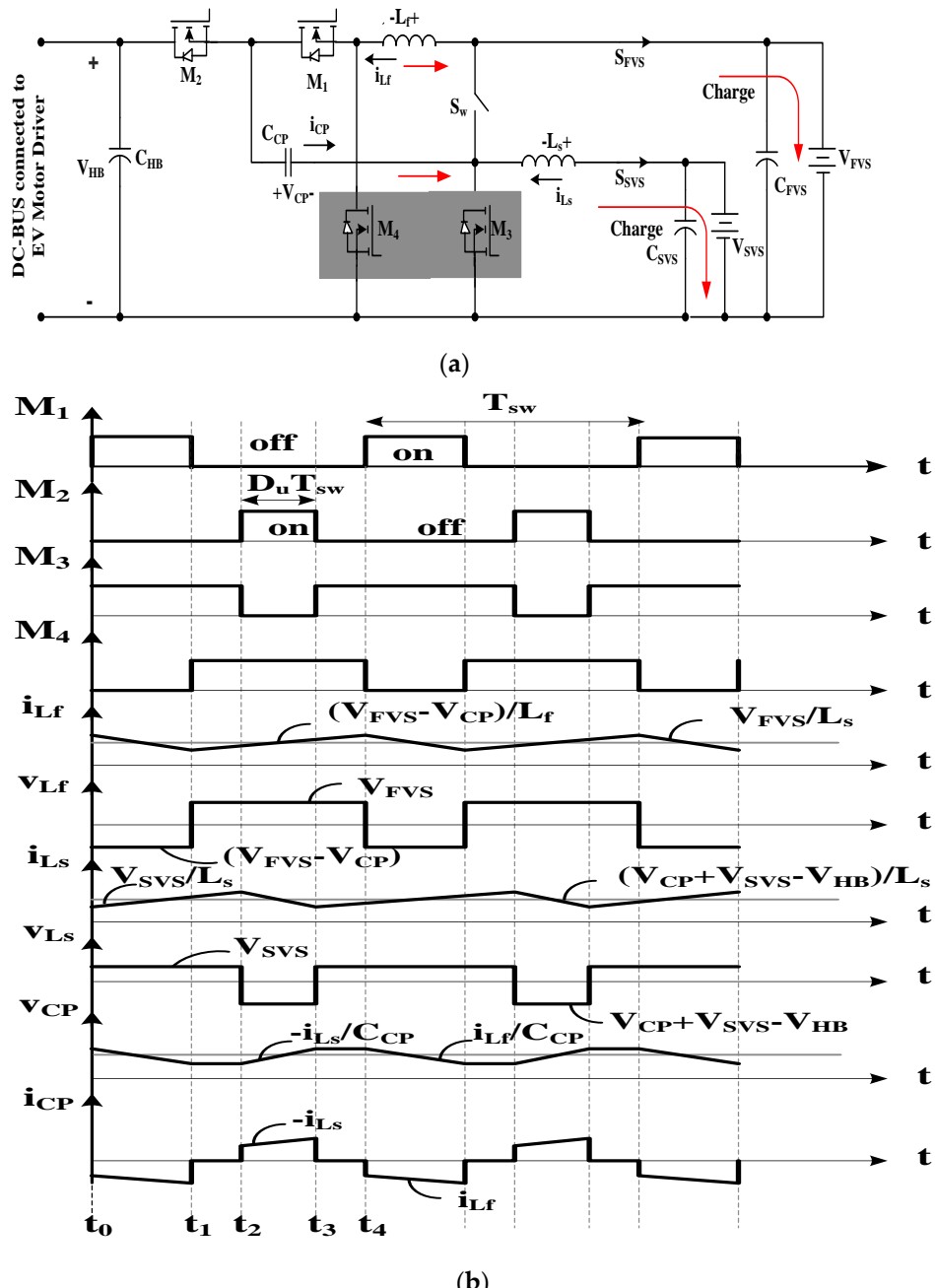

**Figure 5.** (**a**) Schematic Diagram. (**b**) Waveforms of BDC's DC-bus high voltage energy-regenerative mode.

The duty ratio of the energy-regenerating mode for DC-bus high-voltage steady-state waveforms is less than 50%. Their four circuit stages are illustrated through Figure 6a–d. The stage-wise operation is explained in detail below:

(a)  Stage 1 [$t_0 < t < t_1$]: This stage is indicated in Figure 6a. Time interval is $D_t T_s$. Switches $M_1$ and $M_3$ are turned on, whereas switches $M_2$ and $M_4$ are turned off. Voltage across the main inductor, $L_m$, drops linearly from its original value. It is represented by the differential between the charge pump voltage, $V_{CP}$ and the lower side voltage, $V_{FVS}$. An auxiliary inductor, $L_a$ is charged by the energy source $V_{AVS}$. Voltage across the auxiliary inductor, $L_a$ increases linearly. Voltage across the inductors $L_f$ and $L_m$ in stage 1 is denoted by:

$$L_f \frac{di_L f}{dt} = V_{FVS} - V_{CP} \tag{9}$$

$$L_s \frac{di_{Ls}}{dt} = V_{SVS} \tag{10}$$

(b) Stage 2 [$t_1 < t < t_2$]: Time interval in this stage is $(0.5 - D_t)$Ts. $M_3$ and $M_4$ switches are ON, while $M_1$ and $M_2$ switches are OFF, as shown in Figure 6b. Positive lower side voltages $V_{MVS}$ and $V_{AVS}$ are located between the first and second inductors, respectively. These inductor currents increase linearly. Under stage 2, the voltages between the inductors $L_f$ and $L_s$ are indicated by:

$$L_f \frac{di_{Lf}}{dt} = V_{FVS} \tag{11}$$

$$L_s \frac{di_{Ls}}{dt} = V_{SVS} \tag{12}$$

(c) Stage 3 [$t_2 < t < t_3$]: Time interval is $D_t$Ts at this stage. Switches $M_1$ and $M_3$ are off. $M_2$ and $M_4$ switches are on, as shown in Figure 6c. The differential between the lower side voltages $V_{FVS}$ and the charge pump voltages $V_{CP}$ indicates the voltage across the main inductor $L_f$, and the lower side voltage $V_{SVS}$. ɪts level is negative. Under stage 3, the voltages across the inductors $L_f$ and $L_s$ are denoted by:

$$L_f \frac{di_{Lf}}{dt} = V_{FVS} \tag{13}$$

$$L_s \frac{di_{Ls}}{dt} = V_{SVS} + V_{CB} - V_{HB} \tag{14}$$

(d) Stage 4 [$t_3 < t < t_4$]: Time interval at this stage is $(0.5 - D_t)$Ts. $M_3$ and $M_4$ switches are on, while the $M_1$ and $M_2$ switches are off, as shown in Figure 6d. Under stage 4, the voltage across the inductors $L_f$ and $L_s$ is denoted by:

$$L_f \frac{di_{Lf}}{dt} = V_{FVS} \tag{15}$$

$$L_s \frac{di_{Ls}}{dt} = V_{SVS} \tag{16}$$

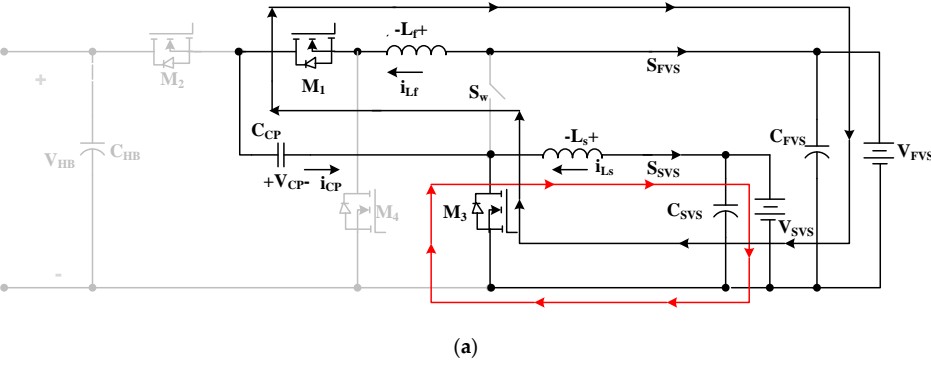

(a)

**Figure 6.** *Cont.*

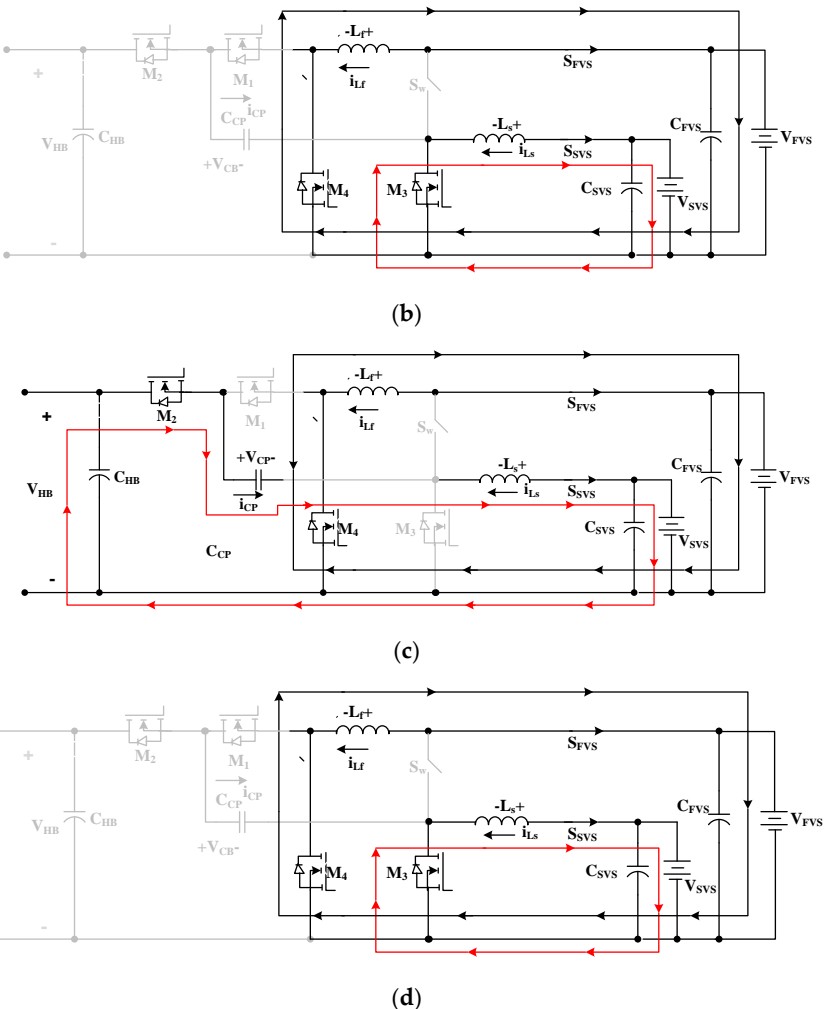

(b)

(c)

(d)

**Figure 6.** Circuit stages in regenerating mode at high voltage.

### 2.3. Buck/Boost Mode with Dual Low Voltage Sources

In this mode, energy stored in the first voltage source is transferred energy to the second voltage source and vice versa, as we can observe in Figure 7a.

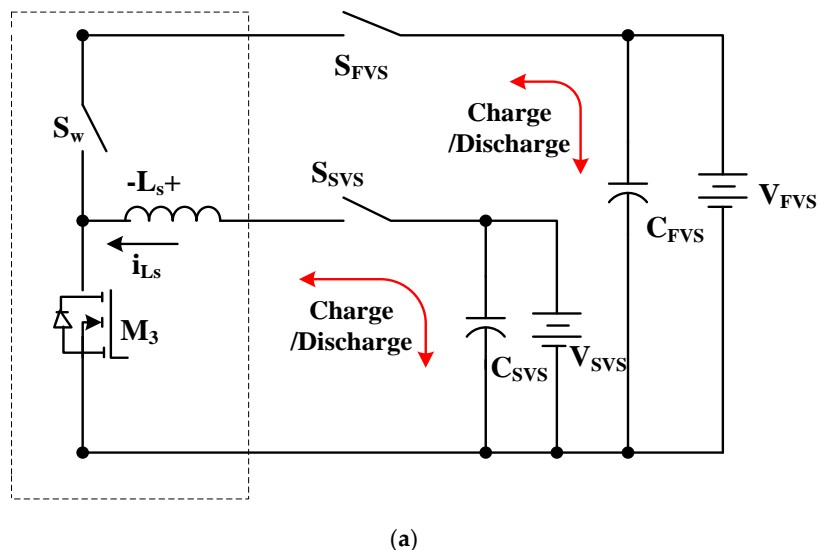

(a)

**Figure 7.** *Cont.*

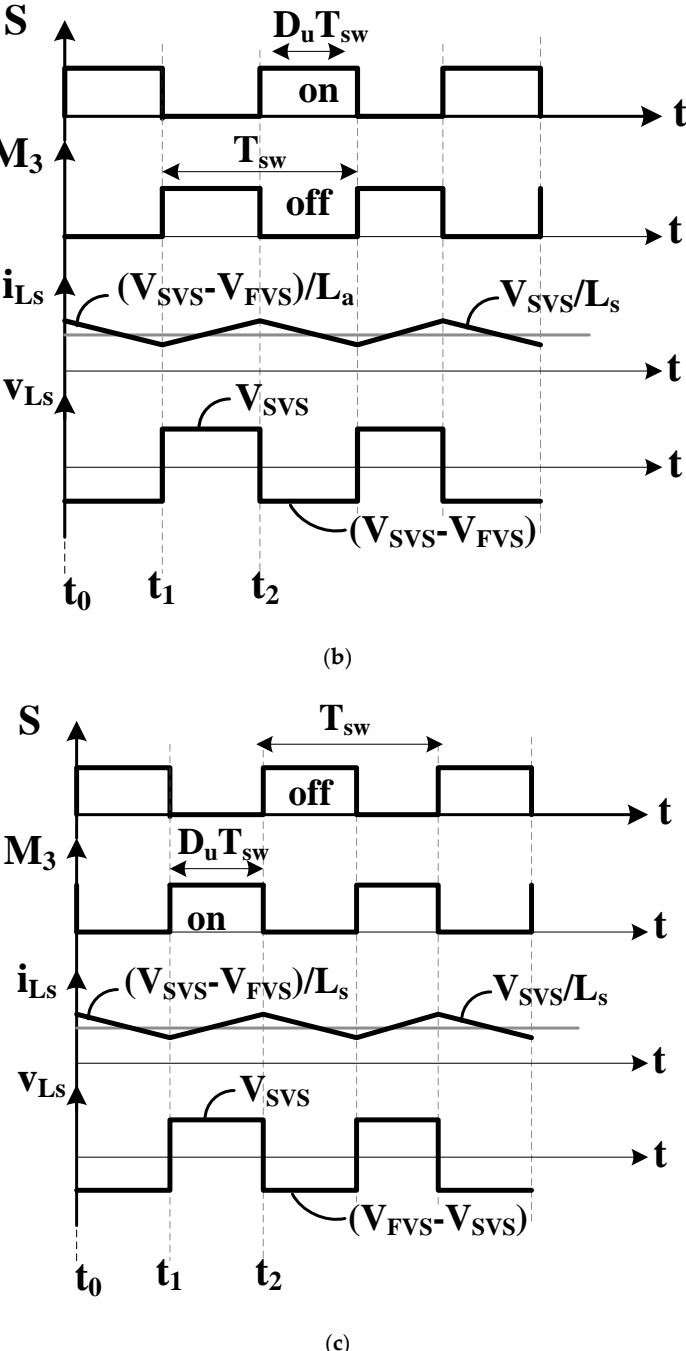

**Figure 7.** BDC's dual low-voltage sources buck/boost mode. (**a**) Schematic Diagram. (**b**) Buck-mode stable-state waveforms. (**c**) Boost-mode stable-state waveforms.

If the duty cycle of active switch $S_w$ is controlled, as shown in Figure 8, it operates as a buck converter. It is because power flows from first voltage source to second voltage source. This depicts the circuit stages in buck mode of the planned BDC with dual low-voltage sources. If the duty cycle of switch $M_3$ is regulated, the converter will operate in boost mode, as shown in Figure 9. It depicts the boost mode of the expected BDC circuit stages for the dual low-voltage sources. In this mode, power transfers from the second voltage source to the first voltage source.

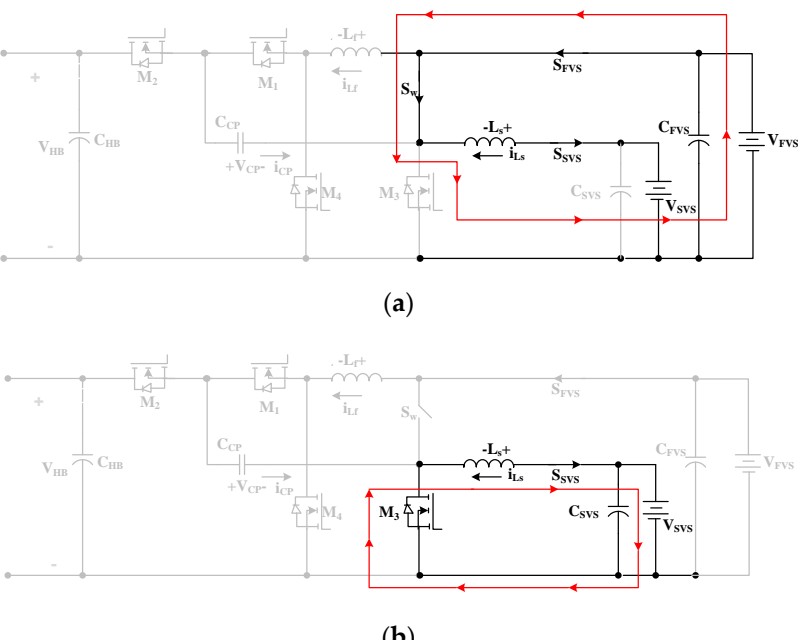

**Figure 8.** (**a**) Stage 1 (**b**) Stage 2.

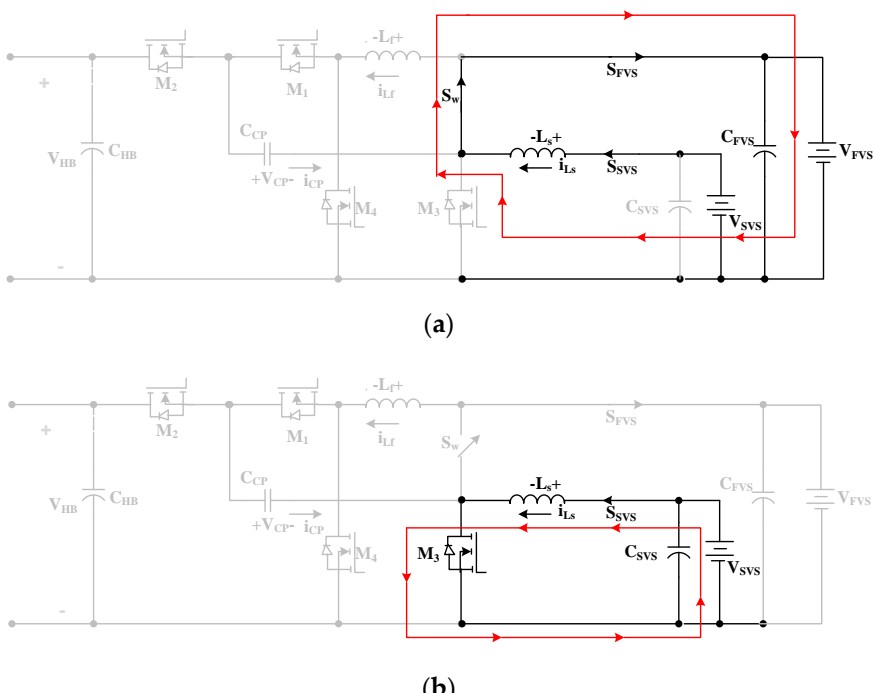

**Figure 9.** (**a**) Stage 1. (**b**) Stage 2.

## 3. Converter Control

Figure 10 shows the block diagram of the feedback-loop control technique, which comprises choosing a vehicle strategic level and mode of BDC to govern pulse width modulation switching schemes. Several operating modes of the BDC controller are depicted in Figure 11. As indicated in the closed loop controller architecture, the vehicular key level necessitates power demand ($P_{Dem}$), vehicle voltage, and power management units. They serve as input signals to BDC mode selection. Overall results of the power management units improve the utilization of voltage sources to meet the power train's power demand, satisfying the requirements of the driver. In Figure 10 inductor currents, $i_{Lm}$ and $i_{La}$ are

calculated and then compared to the reference converter currents. In this mode of controller operation, different driving states of power demand ($P_{Dem}$) and voltage sources ($V_{FVS}$, $V_{SVS}$) are used to determine the vehicle condition. Then, using an artificial neural network, appropriate current references- $i_{LmRef}$ or $i_{LaRef}$ are selected to control the power switches- $S_w$, $M_1$, $M_2$, $M_3$ and $M_4$. Pulse width modulation (PWM) switching methods transform the duty cycle into regulated gate signals for the active switches ($S_w$, $M_1$, $M_4$), and the BDC controller's switch selectors ($S_1$, $S_2$).

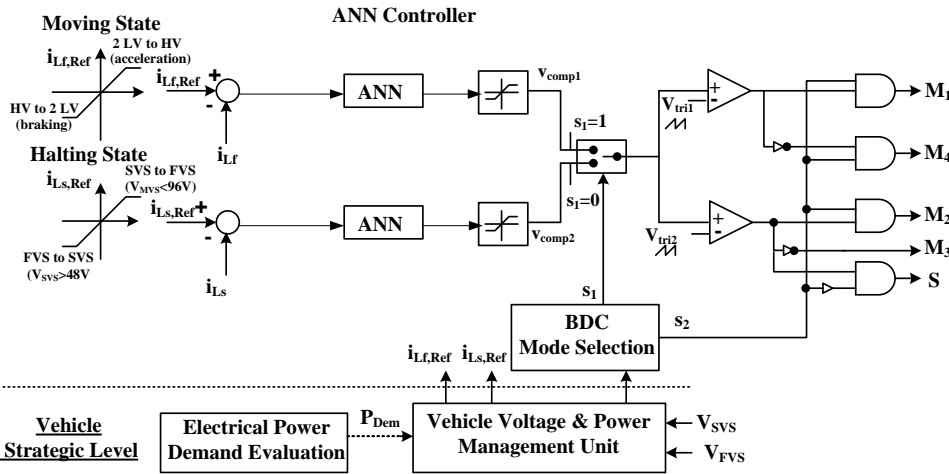

**Figure 10.** Block diagram for feedback loop control technique.

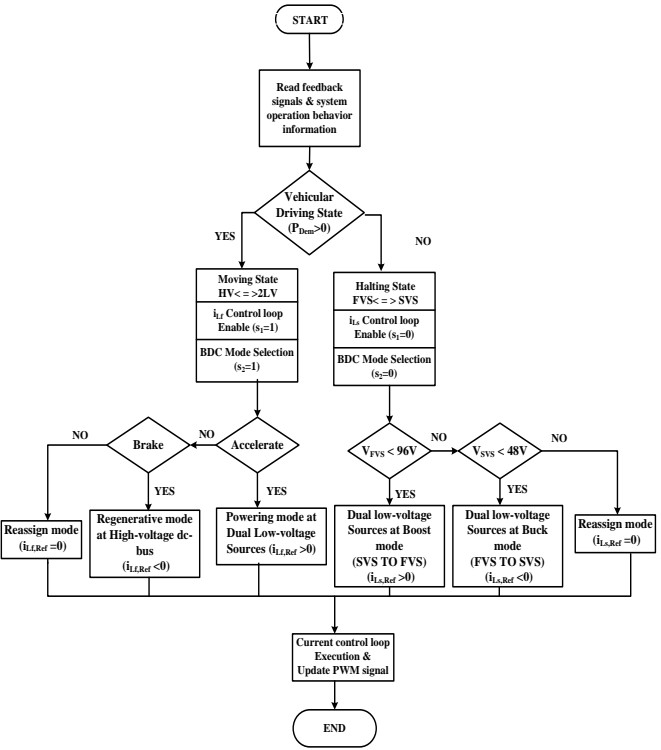

**Figure 11.** Flowchart for the planned BDC's multiple working modes.

$i_{LmRef}$ is used to regulate the power flowing from 2 LV to HV or HV to 2 LV on dual low and high DC-buses voltage sources, whereas $i_{LaRef}$ is also used to regulate the power flowing from the first to the second voltage source and vice versa (i.e., from $V_{FVS}$ to $V_{SVS}$ or from $V_{SVS}$ to $V_{FVS}$).

When the vehicle is in moving state ($P_{Dem} > 0$), the control loop s = 1 in the controller $i_{Lm}$ is activated, and it operates in two modes: accelerating and braking, as seen in the

controlled switches in Table 1. If both conditions are not met, it returns to the reassign mode to operate the very next perception of switching mode. Furthermore, while the vehicle is in a halting condition ($P_{Dem} < 0$), the control loop s1 = 0 in the controller $i_{La}$ will work in two modes, buck and boost, as illustrated in the controlled switches in Table 1. Voltages $V_{FVS}$ (96 V) and $V_{SVS}$ (48 V) are used to perceive mode change in this halting condition. During boost mode, i.e., when $V_{FVS}$ is less than 96 V and $i_{LaRef} > 0$, power flows from the $V_{SVS}$ to the $V_{FVS}$. During buck mode, i.e., when $V_{FVS}$ is less than 46 V and $i_{LaRef} < 0$, power flows from $V_{FVS}$ to the $V_{SVS}$. If neither of these conditions is met, the system reverts to the reassign mode, which is used to process the next perception of mode switching. Figure 12a,b show the various steps in modeling of an ANN controller implemented in this research work.

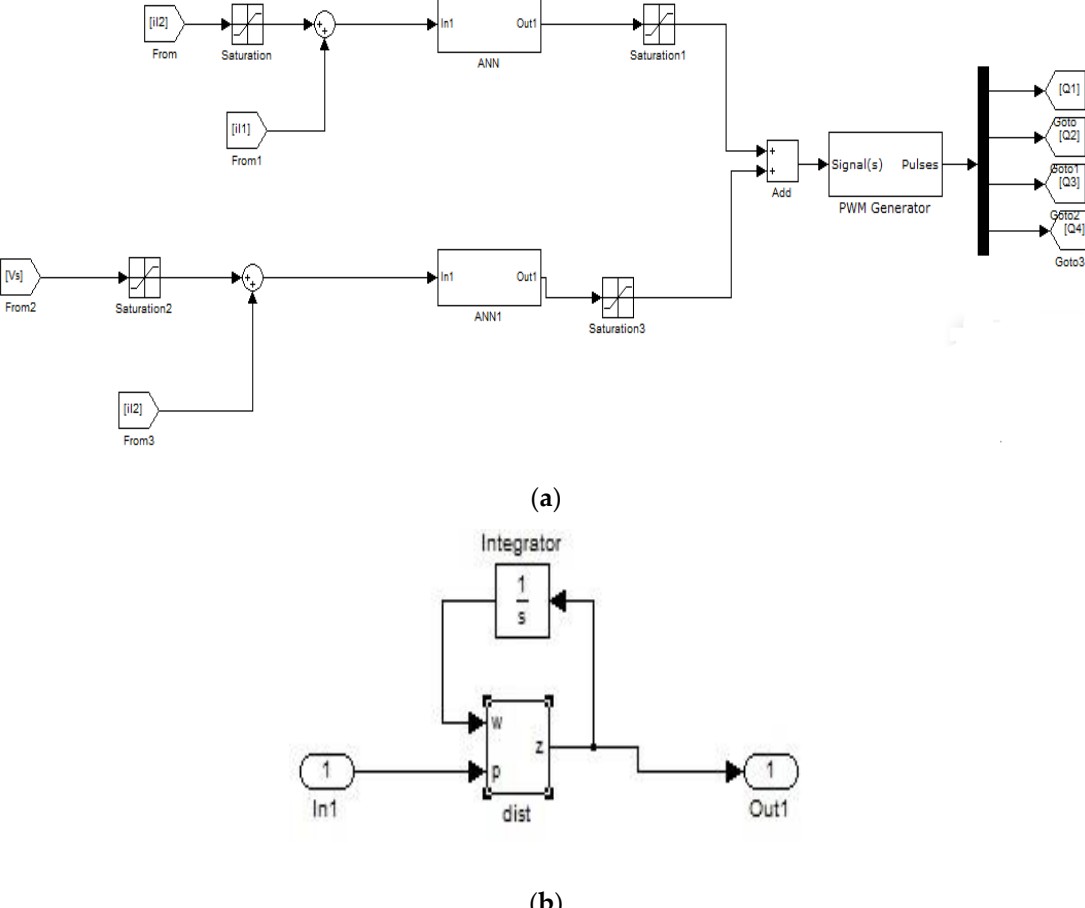

(**a**)

(**b**)

**Figure 12.** (**a**). Switching mechanism of ANN controller. (**b**). Euclidean distance ANN method.

## 4. Discussion

A simulation model is provided in Figure 2 to validate the feasibility and performance of the BDC converter topology with dual battery. It was modelled in MATLAB/Simulink and the system's four operation modes are investigated. Table 2 contains the system parameters.

A traction motor was powered by both sources, which increased the two input voltages, as shown in Figure 14. As a result, the DC-bus voltage $V_{HB}$ of 430 V was influenced by both sources. Performance in powering mode is illustrated in Figures 13 and 14 with performance in regenerative mode shown in Figure 15.

**Table 2.** Parameters and ratings of the HEV system.

| Parameters | Ratings |
|---|---|
| Inductors | $L_f = L_s = 250\ \mu H$ |
| High-side capacitor | $C_{HB} = 1880\ \mu F$ |
| Low-side capacitor | $C_{FVS} = C_{SVS} = 400\ \mu F$ |
| Charge-pump capacitor | $C_{CP} = 10\ \mu F$ |
| ESR of inductance | $R_{Lf} = R_{Ls} = R_L = 50\ m\Omega$ |
| ESR of capacitance | $R_{CB} = 20\ m\Omega,\ R_{FVS} = R_{SVS} = 50\ m\Omega$ |
| Line resistance | $R_{FVS} = 12\ m\Omega,\ R_{SVS} = 6\ m\Omega$ |
| First voltage source | FVS = 96 V |
| Second voltage source | SVS = 40 V |

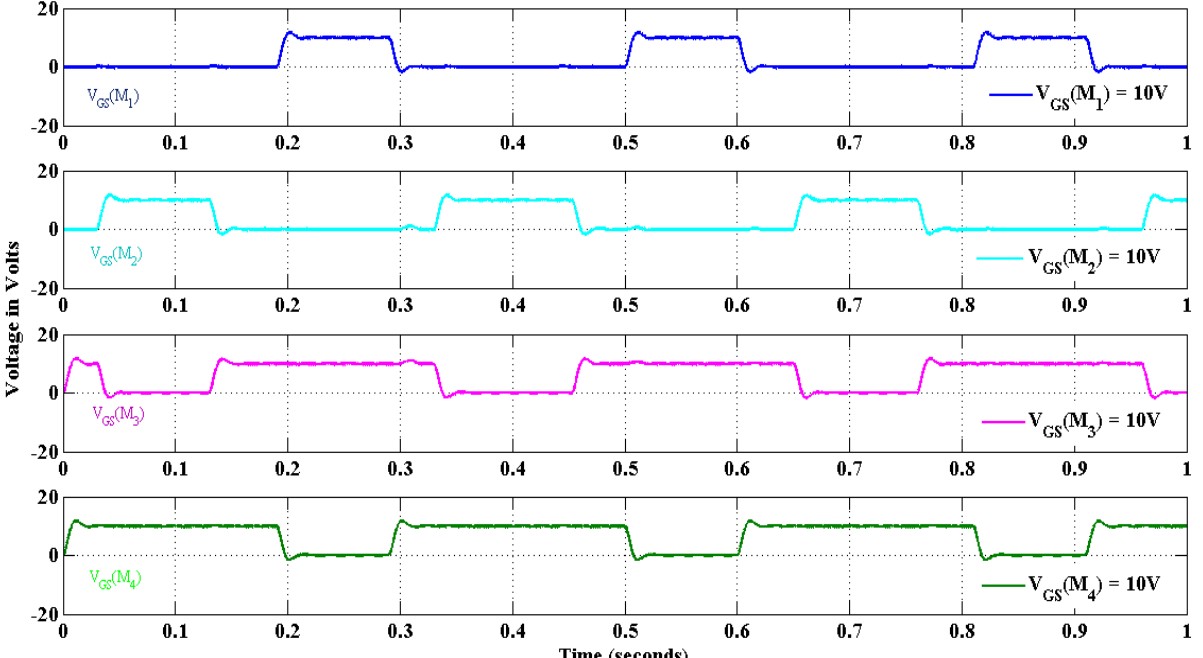

**Figure 13.** Gate signals in powering mode.

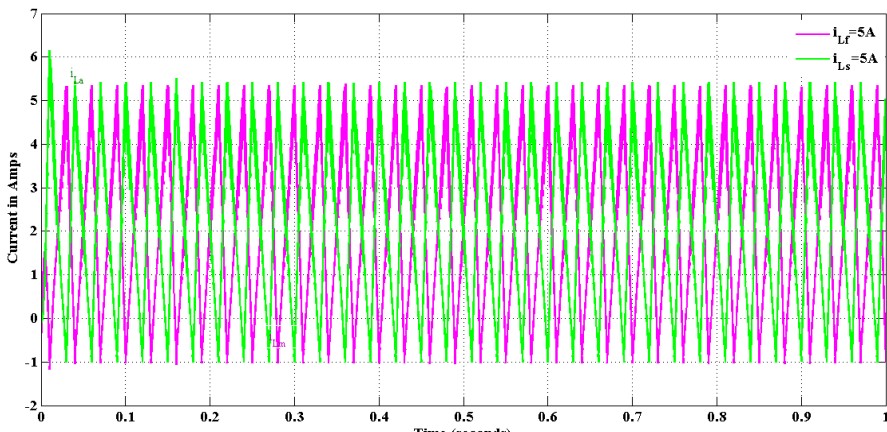

**Figure 14.** Inductor current wave form in powering mode.

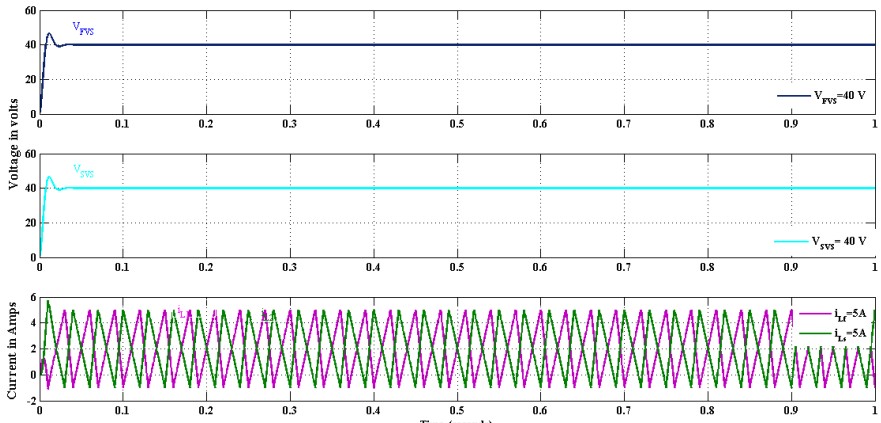

**Figure 15.** Output voltages and inductor currents in regenerative mode.

Figure 13 show the gate pulses delivered to switches $M_1$, $M_2$, $M_3$, and $M_4$, in powering mode and regenerative mode, respectively. Inductor currents $i_{LF}$ and $i_{LS}$ are shown in Figures 14 and 15, respectively. Figure 15 illustrates inductor current and output voltage in DC-bus energy-regenerative mode at high voltage.

Power transfer from the DC-bus towards the main and auxiliary voltage sources is shown in Figure 15. Input power flows in the opposite direction, as clearly postulated from the inductor currents $i_{LF}$ and $i_{LS}$. On the lower side output voltages, $V_{FVS}$ and $V_{SVS}$ were about 96 V and 48 V, respectively.

Using dual low-voltage sources, simulated waveforms–gate signals, inductor currents and output voltages during buck and boost modes are shown in Figures 16 and 17, respectively. In Figure 16, the currents in the inductor are inverted from those in Figure 17. This validates bidirectional power to flow between the second and first voltage sources. Finally, all of the simulation results are consistent with the stable-state prediction study. Waveforms of regulated current step variations for the conceptual design in the high voltage energy-regenerative mode and the dual low-voltage sources-powering mode are displayed in Figures 18 and 19, respectively. In order to transmit energy from the dual lower-side sources to the higher-side DC-bus, the current in the inductor $i_{LF}$ and the higher-side current $i_{HB}$ were adjusted, as illustrated in Figure 18. Negative current output waveforms are indicated in Figure 19. The power flow was effectively inverted.

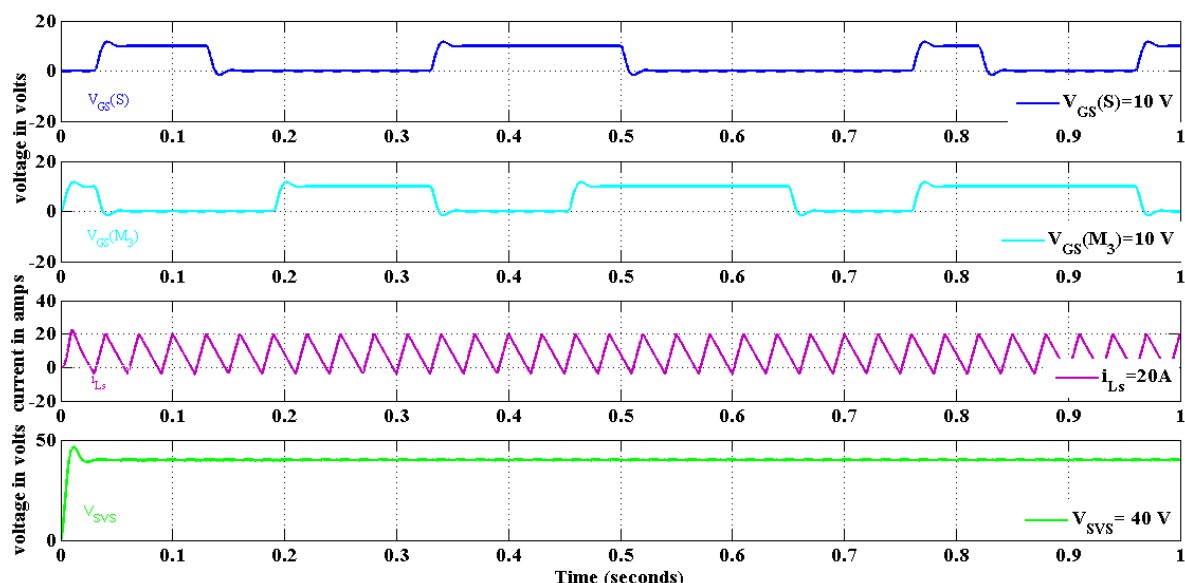

**Figure 16.** Gate signals, output voltage and inductor currents for buck mode.

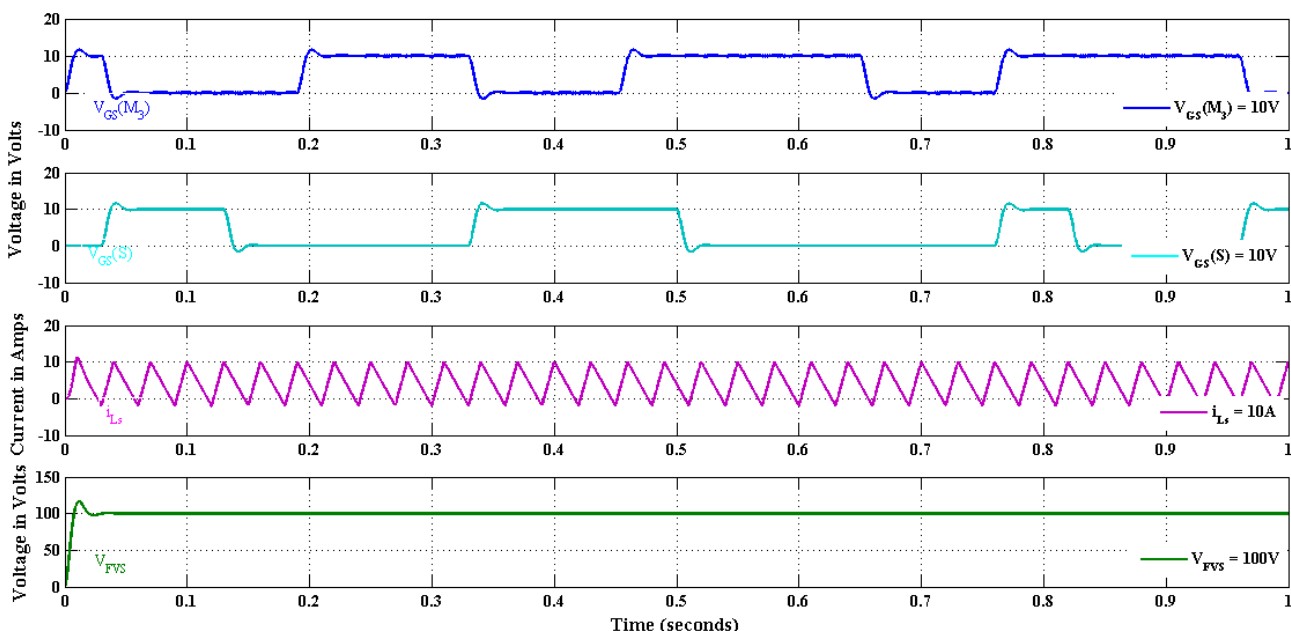

**Figure 17.** Gate signals, output voltage and inductor currents for boost mode.

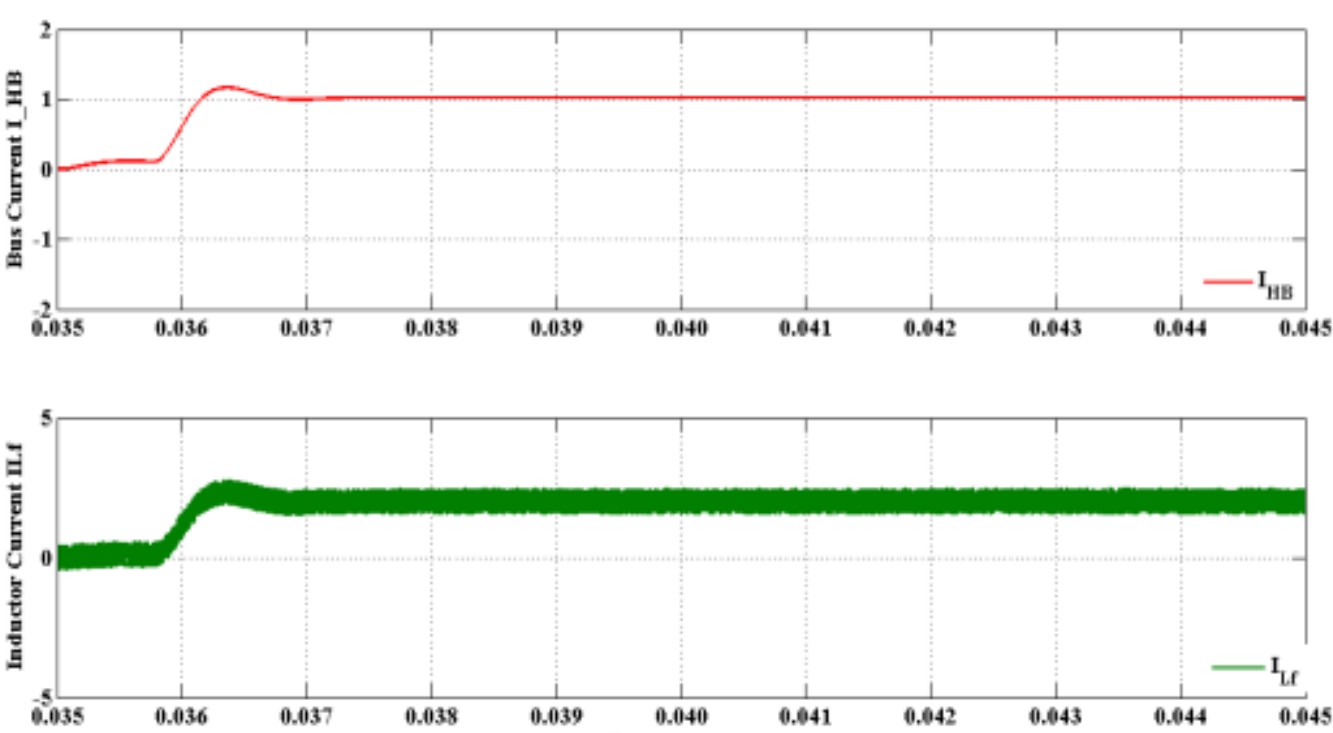

**Figure 18.** Regulated current step variations in inductor currents at dual low voltage sources.

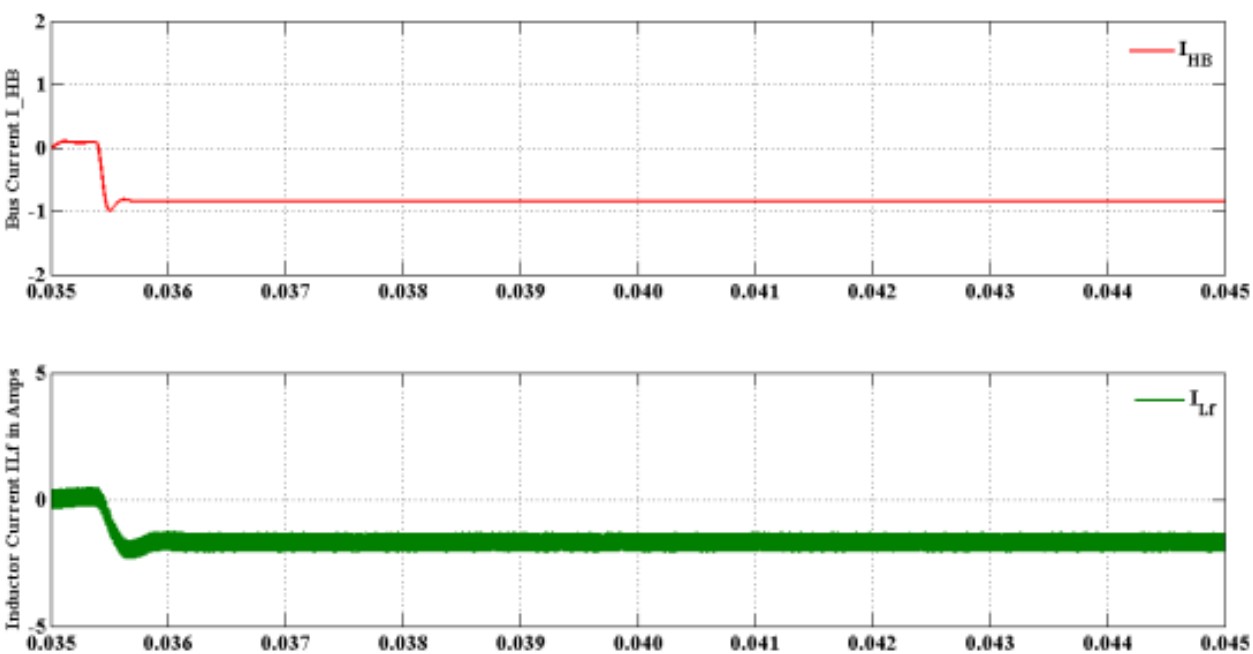

**Figure 19.** Regulated current step variations in inductor currents at high bus voltage.

Figures 20 and 21 show the waveforms of regulated current step variations for the conceptual model for the dual low-voltage sources buck and boost modes, respectively. The higher-side current $i_{FVS}$ and the inductor current $i_{LS}$ were altered to distribute energy between two low-voltage source currents. Inductor current $i_{LS}$ and the first voltage source current $i_{FVS}$ were controlled to transmit the power from the second voltage source to the first voltage source. In Figure 20, the output waveforms of negative current were effectively inverted, and in Figure 21, the power flow was effectively inverted.

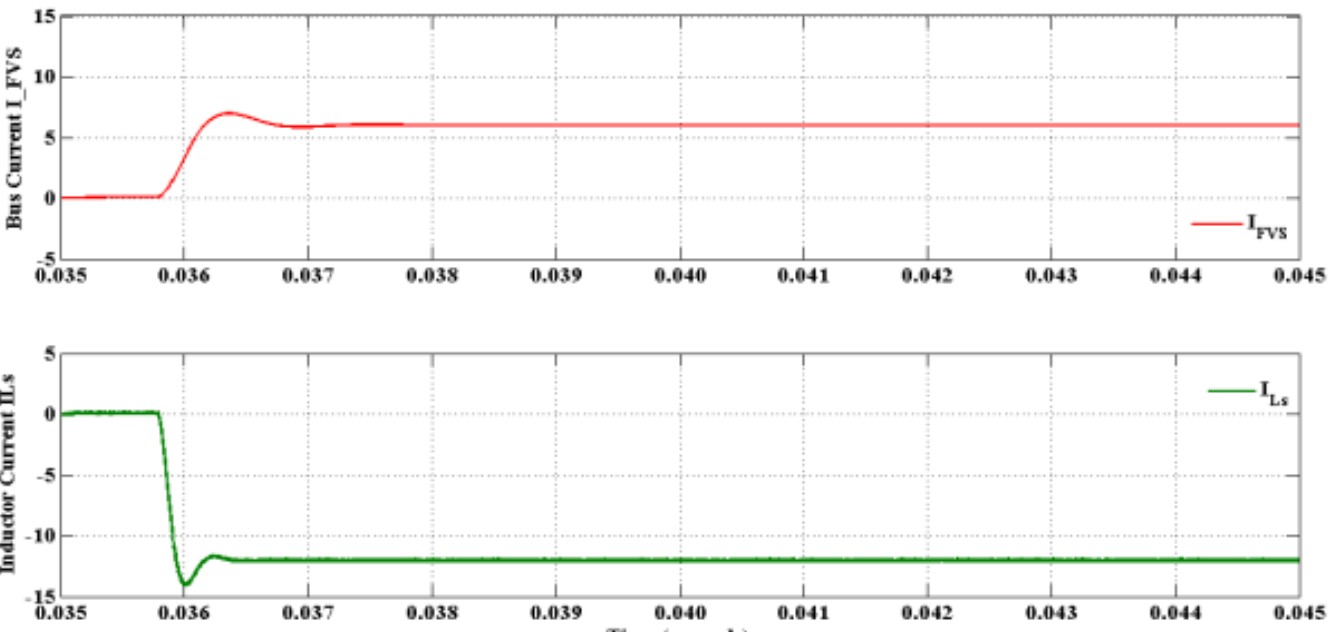

**Figure 20.** Regulated current step variations in inductor currents at buck mode.

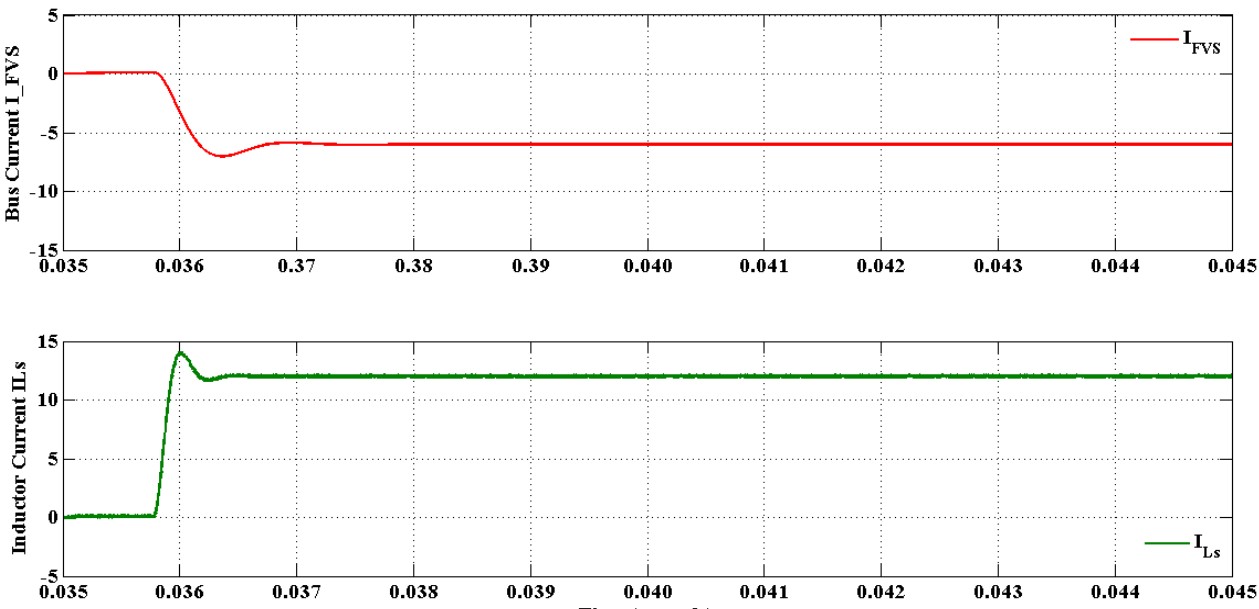

**Figure 21.** Regulated current step variations in inductor currents in boost mode.

*Comparison of PI and ANN Controller Output Waveforms*

Figure 22 shows a comparison of output voltage in power sharing mode at low voltage, while Figure 23 illustrates a comparison of output voltage in DC-bus high-voltage regenerative mode. Figures 24 and 25 show controlled current step variation in inductor currents in buck and boost modes of operation.

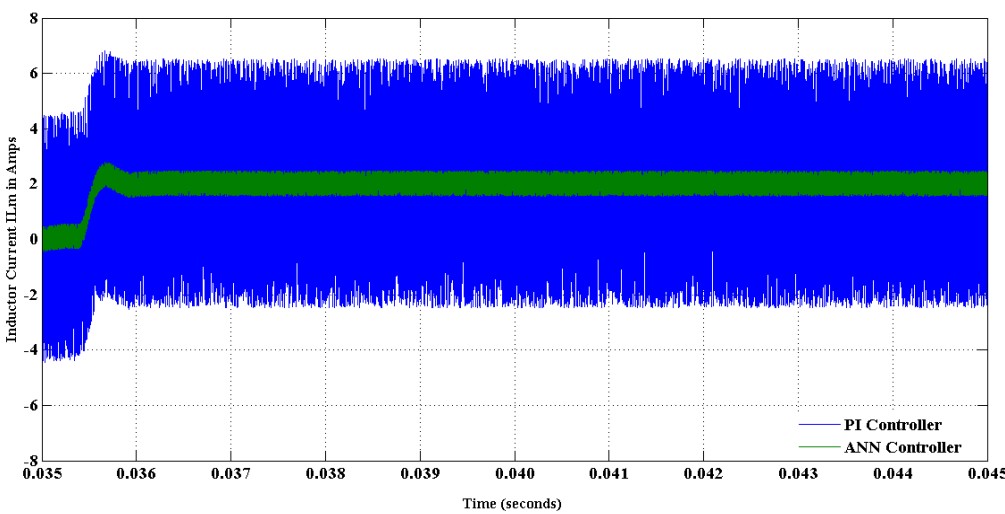

**Figure 22.** Regulated current step variations in inductor currents at dual low voltage sources.

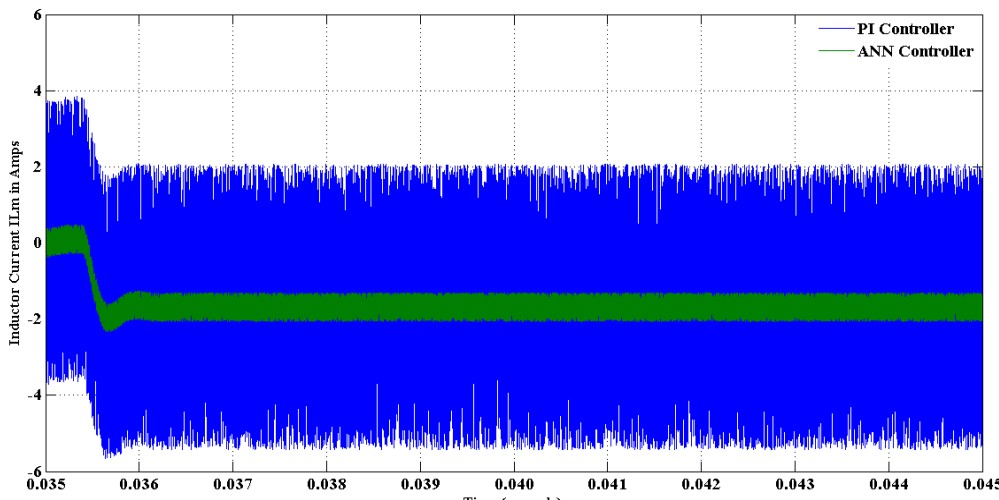

**Figure 23.** Regulated current step variations in inductor currents at high bus voltage.

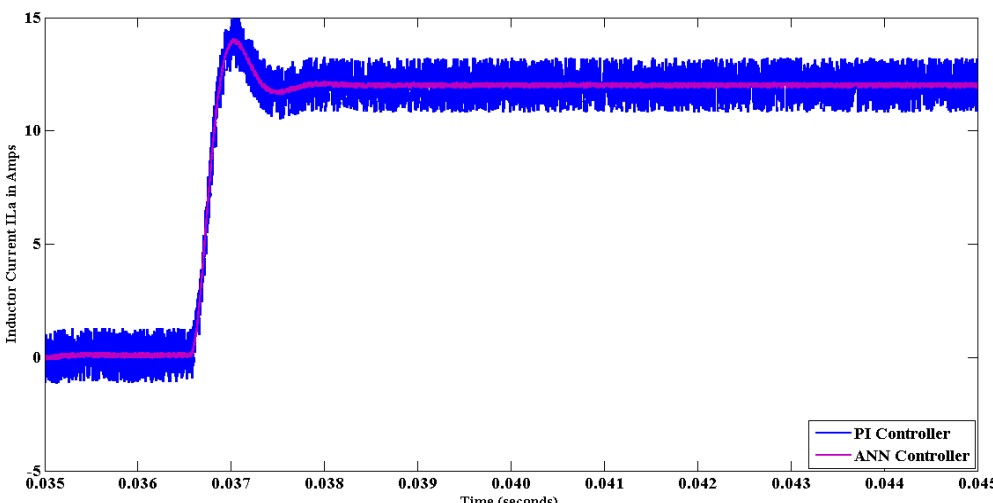

**Figure 24.** Regulated current step variations in inductor currents during boost mode.

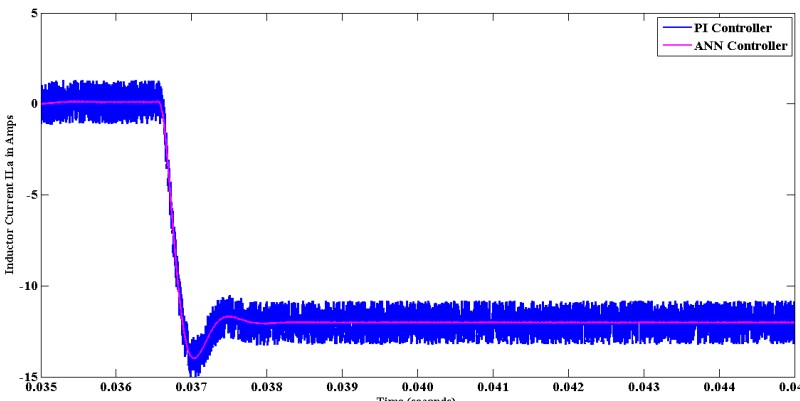

**Figure 25.** Regulated current step variations in inductor currents during buck mode.

Comparative analysis highlights that an ANN controller has a significantly faster rising time and processing speed than the PI controller, and the ANN controller has reduced ripple content in the current signal.

## 5. Conclusions

Dual battery voltage sources were integrated with higher-voltage DC-buses of varying voltage levels using the BDC architecture. Various forms of power transfer were used to discuss the circuit architecture and operational concepts of the BDC. To thoroughly investigate the energy flow between the first and the second voltage sources under the strong variations in voltage levels, regenerative mode, powering mode and inductor currents, a controlled DC–DC voltage static converter with a neural network controller was developed and implemented using MATLAB/Simulink. This controller was simpler to implement, with a limited number of tests carried out in MATLAB. To corroborate the robustness of the proposed converter with the neural network controller, strong variations in voltage levels of the voltage sources, regenerative and powering modes were applied to the hybrid electric vehicle. The simulation results show the effectiveness and the robustness of both the proposed controller and converter to control the power flow between various voltage sources and minimize the current ripples between DC-bus and voltage sources. As a result, inductor current oscillations were reduced.

In future work, efficacy of the proposed BDC should be assessed in a realistic EV domain, employing advanced controllers. Finally, as an obligatory experimentation to this study, the practical implementation of the proposed DC–DC converter should be performed in future research.

**Author Contributions:** R.S.R.S.: Conceptualization, supervision, investigation, methodology, analysis; K.D.K.: writing—original draft preparation, visualization; B.A.: funding acquisition, data curation; M.A.: resources, project administration. All authors have read and agreed to the published version of the manuscript.

**Funding:** This research was funded from Taif University Researchers Supporting Project Number (TURSP-2020/278), Taif University, Taif, Saudi Arabia.

**Data Availability Statement:** Not applicable.

**Acknowledgments:** The authors would like to acknowledge the financial support received from Taif University Researchers Supporting Project Number (TURSP-2020/278), Taif University, Taif, Saudi Arabia.

**Conflicts of Interest:** The authors declare that they have no known competing financial interest or personal relationships that could have appeared to influence the work reported in this paper.

## Notations

Notations used throughout the paper are stated below:

| | |
|---|---|
| $V_{HB}$ | DC-bus high voltage |
| $V_{FVS}$, $V_{SVS}$ | Dual low-voltage sources |
| $C_{CP}$ | Charge pump capacitor |
| $M_1$, $M_2$, $M_3$, $M_4$ | Four active switches |
| $L_f$, $L_s$ | Phase inductors |
| $i_{Lf}$ | Main phase inductor current |
| $i_{Ls}$ | Auxiliary phase inductor current |
| $S_W$, $S_{FVS}$, $S_{SVS}$ | Three bidirectional switches |
| $i_{CP}$ | Charge pump current |
| $C_{FVS}$ | First voltage source capacitor |
| $C_{SVS}$ | Second voltage source capacitor |

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
