# Peer review of "A Smart ANN-Based Converter for Efficient Bidirectional Power Flow in Hybrid Electric Vehicles"

_electronics, doi:10.3390/electronics11213564_

Round 1

Reviewer 1 Report

The topic A Smart ANN Based Converter for Efficient Bidirectional Power Flow in Hybrid Electric Vehicles is interesting, however, there are some issues that should be addressed by the authors: The Introduction" sections can be made much more impressive by highlighting your contributions. The contribution of the study should be explained simply and clearly. The authors should further enlarge the Introduction with current work about advanced control and optimization algorithms to improve the research background, for example, optimal design of low computational burden model predictive control based on SSDA towards autonomous vehicle under vision dynamics; Robust Model Predictive Control Paradigm for Automatic Voltage Regulators against Uncertainty Based on Optimization Algorithms‏‏.

Clarify how you handle the constraints of the model during the analysis.

Clarify the practical implementation of the proposed method

Clarify how you adjust the parameters of the proposed ANN

Increase the resolution of the figures

Compare your method with other in literature

Conclusion section should be rearranged. According to the topic of the paper, the authors may propose some interesting problems as future work in the conclusion.

This study may be proposed for publication if it is addressed in the specified problems.

Author Response

Manuscript Title: A Smart ANN Based Converter for Efficient Bidirectional Power Flow in Hybrid Electric Vehicles

Subject: Reply to the editorial remarks and reviewers’ comments regarding Manuscript ID: electronics-1926572

Dear Sir,

Greetings of the Day!

The authors express their sincere acknowledgements and thanks to the honourable editor and worthy reviewers for devoting their valuable time to review the manuscript and providing necessary comments and suggestions. These suggestions and comments have undoubtedly contributed to improvement in the overall quality of the paper. The authors have taken the comments very positively and have tried to answer each comment in the best possible way. The comments and suggestions of the honourable editor and reviewers have been incorporated in the revised manuscript and are highlighted using yellow text.

  1. The topic A Smart ANN Based Converter for Efficient Bidirectional Power Flow in Hybrid Electric Vehicles is interesting, however, there are some issues that should be addressed by the authors: The Introduction" sections can be made much more impressive by highlighting your contributions. The contribution of the study should be explained simply and clearly. The authors should further enlarge the Introduction with current work about advanced control and optimization algorithms to improve the research background, for example, optimal design of low computational burden model predictive control based on SSDA towards autonomous vehicle under vision dynamics; Robust Model Predictive Control Paradigm for Automatic Voltage Regulators against Uncertainty Based on Optimization Algorithms‏‏.

Response: We the authors would like to thank honourable reviewer for the suggestion made. The “Introduction” section is rewritten to make objective of the present study more clear and extending more with the literature survey. The interesting results are conveyed in an impressive manner. Authors contributions are included in abstract giving a simple and clear explanation. These are further outlined at the end of Introduction Section. It is highlighted in yellow colour. Thank you.

The main objective of the dc-dc converter is to adjust the output of the front-end ac-dc converter and to charge the EV in desired mode (CC or CV). The most common dc-dc converter topologies include voltage-fed bridges; current-fed bridges; appropriate combinations of these; and resonant converters [13][14].  Number of active switches and thereby device stress is reduced in Dual active voltage-fed full bridges in comparison to Voltage- and current-fed full-bridges. Unidirectional DC-DC converters were studied in [15] for power decoupling between Fuel cell and DC bus.

Considering the diverse vehicle driving settings, widespread voltage matching and decoupling of real and reactive power are crucial. These can be effectively ensured with the bidirectional DC-DC converter. Drawbacks in operation of conventional BDC are reported in the literature [16][17] from different viewpoints, although this paper does not aim to address them. Previous studies proposed several isolated and non-isolated bi-directional topologies to improve dynamic performance, gain, efficiency, and operability of BDCs for energy storage and renewable applications. In [18] bidirectional DC-DC converter for power decoupling in distribution system with PV system and Electric Springs was investigated. A bidirectional dc-dc converter used to regulate the charging current in a bidirectional EV charger is discussed in [19][20]. These operate with isolated and non-isolated circuit arrangements. Another major benefit is reduction in volume, weight, and cost of charger. In this context, several topologies of bidirectional converters specifically applied to electric vehicles have been investigated and a re-viewed [21]. A two-phase interleaved bidirectional DC/DC converter was studied in [22]. A circuit configuration with the aim of increased voltage conversion ratio was proposed in [23]. In [24], a dual active bridge bi-directional converter for enhanced power range for ultracapacitor was designed. Multi-port concept for bidirectional power converter with battery/supercapacitor was extensively simulated in [25]. A bidirectional DC/DC Converter with dual-battery energy storage for Hybrid Electric Vehicle System was developed in [26].

Main objectives of the work is coordinated control of the DC energy sources of various voltage levels, independent power flow between both the energy sources, and regulation of current flow from the dc bus to the voltage sources. This work adopts to optimise the converter control to investigate under diverse combinations of the voltage levels of the sources, energy flow between the sources, modes of operation, inductor currents. The advantage of neural network controller is its simplicity, with limited number of tests to construct it.

A neural network controller was employed to control the interleaved boost DC-DC converter associated to proton exchange membrane fuel cell in [27]. In [28], neural network was implemented in energy management system in electric vehicles using ultra capacitors. Fuzzy neural network PID control was developed in [29] in the pressure control of the EVs. And in [30], Wang et al have proposed a method using Back Propagation Neural Networks in estimation of State of Health of battery in Electric Vehicles.

In this paper, the objectives of the proposed bidirectional DC-DC converter and its controller are regulation of the energy flow between voltage sources as well as the mitigation of the ripples in inductor currents. The proposed solution generates the required duty cycle for this purpose. The proposed PWM allows minimizing the ripple of current for all voltage levels. This paper is organized in such a way that in section 2, the architecture and operating modes of the Dual Converter are elaborated. The control technique of the converter is detailed in section 3. In Section 4, the validation of the proposed vehicle is projected with simulation results. Finally comparison of PI and ANN are presented in section 5.

  1. Clarify how you handle the constraints of the model during the analysis.

Response: We the authors would like to thank honourable reviewer for the suggestion made. The step by step approach during the analysis is elaborated using a flowchart demonstrated in Figure 12.

  1. Clarify the practical implementation of the proposed method.

Response: We the authors would like to thank honourable reviewer for the suggestion made. As a necessary complement to this study, the experimental verification of the proposed bidirectional DC-DC Converter to be performed in future research. And it is out of the scope of the work in this article.

  1. Clarify how you adjust the parameters of the proposed ANN.

Response: We the authors would like to thank honourable reviewer for the suggestion made. Implementation of the proposed ANN in Matlab Simulink is explained in Figure 11 (i), (ii). Switching mechanism of ANN is shown in Figure 11 (i) whereas calculation of the Euclidean distance in ANN method is illustrated in Figure 11 (ii).

  1. Increase the resolution of the figures

Response: We the authors would like to thank honourable reviewer for the suggestion made. Resolution of the figures are increased and included in the revised manuscript.

  1. Compare your method with other in literature.

Response: We the authors would like to thank honourable reviewer for the suggestion made. To corroborate the robustness of the proposed converter with the neural network controller, strong variations in voltage levels of the voltage sources, regenerative and powering modes are applied to the Hybrid Electric Vehicle. The simulation results show the effectiveness and the robustness of the both the proposed controller and converter to control the power flow between various voltage sources and minimize the current ripples between DC bus and voltage sources. As a result of that, inductor current oscillations are reduced.

  1. Conclusion section should be rearranged. According to the topic of the paper, the authors may propose some interesting problems as future work in the conclusion.

Response: We the authors would like to thank honourable reviewer for the suggestion made. Conclusion section is rewritten. Some of the problems are mentioned as future work in the conclusion. These are given below:

Dual battery voltage sources were integrated with higher-voltage DC buses of varying voltage levels using the BDC architecture. Various forms of power transfer were used to discuss the circuit architecture and operational concepts of the BDC. To thoroughly investigate the energy flow between the first and the second voltage sources under the strong variations in voltage levels, regenerative mode, powering mode and inductor currents, a controlled DC-DC voltage static converter with neural network controller is developed and implemented using Matlab-Simulink. This controller is simpler to implement with limited number of tests carried out in Matlab. To corroborate the robustness of the proposed converter with the neural network controller, strong variations in voltage levels of the voltage sources, regenerative and powering modes are applied to the Hybrid Electric Vehicle. The simulation results show the effectiveness and the robustness of the both the proposed controller and converter to control the power flow between various voltage sources and minimize the current ripples between DC bus and voltage sources. As a result of that, inductor current oscillations are reduced.

In future work, efficacy of the proposed BDC should be assessed in a realistic EV domain, employing advanced controllers. Finally, as an obligatory experimentation to this study, the practical implementation of the proposed DC-DC converter should be performed in future research.

Reviewer 2 Report

In comparison with the following paper, There is no novelty in this manuscript:

Sai Teja, Tatoju, M. D. Yaseen, and T. Anilkumar. "Bidirectional DC to DC converter with ANN controller for hybrid electric vehicle." Int. J. Innov. Technol. Explor. Eng 8 (2019): 4446-4453.

Author Response

We the authors would like to thank honourable reviewer for the suggestion made. Main objectives of the work is coordinated control of the DC energy sources of various voltage levels, independent power flow between both the energy sources, and regulation of current flow from the dc bus to the voltage sources. This work adopts to optimise the converter control to investigate under diverse combinations of the voltage levels of the sources, energy flow between the sources, modes of operation, inductor currents. The advantage of neural network controller is its simplicity, with limited number of tests to construct it. In this paper, the objectives of the proposed bidirectional DC-DC converter and its controller are regulation of the energy flow between voltage sources as well as the mitigation of the ripples in inductor currents. A step by step approach during the extensive analysis is elaborated using a flowchart demonstrated in Figure 12. Extensive simulations are carried out for various voltage levels and The proposed solution generates the required duty cycle for this purpose. The proposed PWM minimized the ripple of current for all voltage levels as illustrated in Figure 19-26. 

Reviewer 3 Report

the article called "A Smart ANN Based Converter for Efficient Bidirectional Power Flow in Hybrid Electric Vehicles" is interesting but should be developed better.

I find that the introduction does not describe other articles that develop control between two different voltage architectures. It would be appropriate to report some similar articles to make a comparison.

Paragraph 2.1 is clear to me, I would like you to discuss the power, in percentage, how many power can be drawn from the higher voltage system and how much from the lower one, since the two systems are unable to work at the same time.

If possible the energy flow of the higher voltage system, I would put it in blue color and not black.

Paragraph 2.2 left me with the doubt that the regenerative flux does not go to the higher voltage system, as shown in figure 5.i, directly, but is only obtainable from the lower voltage system (5.i) and I wonder if taking this energy is the first phase in regenerative braking. More descriptions need to be added in the text.

If possible the energy flow of the higher voltage system, I would put it in blue color and not black.

I would also like to suggest that the formulas should be aligned.

The control part is flowing.

The discussion, I would have a doubt about the figure 2 mentioned, perhaps it is a mistake.

I have difficulty reading the axes from figure 13 onwards, I would suggest using larger fonts.

The conclusions must be expanded, the numerical results must be recalled and comparisons must be made.

Author Response

Dear Sir,

Greetings of the Day!

The authors express their sincere acknowledgements and thanks to the honourable editor and worthy reviewers for devoting their valuable time to review the manuscript and providing necessary comments and suggestions. These suggestions and comments have undoubtedly contributed to improvement in the overall quality of the paper. The authors have taken the comments very positively and have tried to answer each comment in the best possible way. The comments and suggestions of the honourable editor and reviewers have been incorporated in the revised manuscript and are highlighted using yellow text.

  1. The article called "A Smart ANN Based Converter for Efficient Bidirectional Power Flow in Hybrid Electric Vehicles" is interesting but should be developed better. I find that the introduction does not describe other articles that develop control between two different voltage architectures. It would be appropriate to report some similar articles to make a comparison.

Response: We the authors would like to thank honourable reviewer for the suggestion made. The “Introduction” section is rewritten to make objective of the present study more clear and extending more with the literature survey. The interesting results are conveyed in an impressive manner. Authors contributions are included in abstract giving a simple and clear explanation. These are further outlined at the end of Introduction Section. It is highlighted in yellow colour. Thank you.

The main objective of the dc-dc converter is to adjust the output of the front-end ac-dc converter and to charge the EV in desired mode (CC or CV). The most common dc-dc converter topologies include voltage-fed bridges; current-fed bridges; appropriate combinations of these; and resonant converters [13][14].  Number of active switches and thereby device stress is reduced in Dual active voltage-fed full bridges in comparison to Voltage- and current-fed full-bridges. Unidirectional DC-DC converters were studied in [15] for power decoupling between Fuel cell and DC bus.

Considering the diverse vehicle driving settings, widespread voltage matching and decoupling of real and reactive power are crucial. These can be effectively ensured with the bidirectional DC-DC converter. Drawbacks in operation of conventional BDC are reported in the literature [16][17] from different viewpoints, although this paper does not aim to address them. Previous studies proposed several isolated and non-isolated bi-directional topologies to improve dynamic performance, gain, efficiency, and operability of BDCs for energy storage and renewable applications. In [18] bidirectional DC-DC converter for power decoupling in distribution system with PV system and Electric Springs was investigated. A bidirectional dc-dc converter used to regulate the charging current in a bidirectional EV charger is discussed in [19][20]. These operate with isolated and non-isolated circuit arrangements. Another major benefit is reduction in volume, weight, and cost of charger. In this context, several topologies of bidirectional converters specifically applied to electric vehicles have been investigated and a re-viewed [21]. A two-phase interleaved bidirectional DC/DC converter was studied in [22]. A circuit configuration with the aim of increased voltage conversion ratio was proposed in [23]. In [24], a dual active bridge bi-directional converter for enhanced power range for ultracapacitor was designed. Multi-port concept for bidirectional power converter with battery/supercapacitor was extensively simulated in [25]. A bidirectional DC/DC Converter with dual-battery energy storage for Hybrid Electric Vehicle System was developed in [26].

Main objectives of the work is coordinated control of the DC energy sources of various voltage levels, independent power flow between both the energy sources, and regulation of current flow from the dc bus to the voltage sources. This work adopts to optimise the converter control to investigate under diverse combinations of the voltage levels of the sources, energy flow between the sources, modes of operation, inductor currents. The advantage of neural network controller is its simplicity, with limited number of tests to construct it.

A neural network controller was employed to control the interleaved boost DC-DC converter associated to proton exchange membrane fuel cell in [27]. In [28], neural network was implemented in energy management system in electric vehicles using ultra capacitors. Fuzzy neural network PID control was developed in [29] in the pressure control of the EVs. And in [30], Wang et al have proposed a method using Back Propagation Neural Networks in estimation of State of Health of battery in Electric Vehicles.

In this paper, the objectives of the proposed bidirectional DC-DC converter and its controller are regulation of the energy flow between voltage sources as well as the mitigation of the ripples in inductor currents. The proposed solution generates the required duty cycle for this purpose. The proposed PWM allows minimizing the ripple of current for all voltage levels. This paper is organized in such a way that in section 2, the architecture and operating modes of the Dual Converter are elaborated. The control technique of the converter is detailed in section 3. In Section 4, the validation of the proposed vehicle is projected with simulation results. Finally comparison of PI and ANN are presented in section 5.

  1. Paragraph 2.1 is clear to me, I would like you to discuss the power, in percentage, how many power can be drawn from the higher voltage system and how much from the lower one, since the two systems are unable to work at the same time.

Paragraph 2.2 left me with the doubt that the regenerative flux does not go to the higher voltage system, as shown in figure 5.i, directly, but is only obtainable from the lower voltage system (5.i) and I wonder if taking this energy is the first phase in regenerative braking. More descriptions need to be added in the text. If possible the energy flow of the higher voltage system, I would put it in blue color and not black.

Response: We the authors would like to thank honourable reviewer for the suggestion made. In this work, Power transfer from the dc-bus towards the main and auxiliary voltage sources are investigated in terms of output voltage levels and currents. These are extensively studied in regenerative mode, buck mode and boost mode of operations. The same are illustrated in Figure. 16,17, 18 respectively.

  1. I would also like to suggest that the formulas should be aligned.

Response: We the authors would like to thank honourable reviewer for the suggestion made. The formulas are aligned in the revised manuscript.

  1. The discussion, I would have a doubt about the figure 2 mentioned, perhaps it is a mistake.

Response: We the authors would like to thank honourable reviewer for the suggestion made. Figure 2 is thoroughly verified and presented in the manuscript.

  1. I have difficulty reading the axes from figure 13 onwards, I would suggest using larger fonts.

Response: We the authors would like to thank honourable reviewer for the suggestion made. Font sizes of the figures are increased and included in the revised manuscript.

  1. The conclusions must be expanded, the numerical results must be recalled and comparisons must be made.

Response: We the authors would like to thank honourable reviewer for the suggestion made. Conclusion section is rewritten. Some of the problems are mentioned as future work in the conclusion. These are given below:

Dual battery voltage sources were integrated with higher-voltage DC buses of varying voltage levels using the BDC architecture. Various forms of power transfer were used to discuss the circuit architecture and operational concepts of the BDC. To thoroughly investigate the energy flow between the first and the second voltage sources under the strong variations in voltage levels, regenerative mode, powering mode and inductor currents, a controlled DC-DC voltage static converter with neural network controller is developed and implemented using Matlab-Simulink. This controller is simpler to implement with limited number of tests carried out in Matlab. To corroborate the robustness of the proposed converter with the neural network controller, strong variations in voltage levels of the voltage sources, regenerative and powering modes are applied to the Hybrid Electric Vehicle. The simulation results show the effectiveness and the robustness of the both the proposed controller and converter to control the power flow between various voltage sources and minimize the current ripples between DC bus and voltage sources. As a result of that, inductor current oscillations are reduced.

In future work, efficacy of the proposed BDC should be assessed in a realistic EV domain, employing advanced controllers. Finally, as an obligatory experimentation to this study, the practical implementation of the proposed DC-DC converter should be performed in future research.

Reviewer 4 Report

The authors of this manuscript propose to work on an ANN-based smart converter for efficient bidirectional power flow in hybrid electric vehicles. The main objectives of this work are coordinated control of DC power sources of different voltage levels, independent power flow between the two power sources, and regulation of current flow from the DC bus to the voltage sources.

Although the proposed topic is related to electronics topics, the manuscript as it stands has many flaws that the authors need to correct.

I therefore propose the following major revisions.

1) The abstract needs to be reworked as the authors' contributions are not clearly expressed. In addition, the main ins and outs need to be detailed and quantified.

2) The authors did not conduct an exhaustive review of the literature to show the significance of their work in relation to what is already available in the literature. Please complete this literature synthesis to identify research questions. Authors should cite recent work from the MDPI group journals (Electronics, Energies, ...).

3) The manuscript as it stands has far too many sections. The authors should organize their work in a more traditional way: 1. Introduction; 2. Related Works; 3. Materials and Methods; 4. Main Results; 5. Discussion; 6. Conclusions.

4) I regret that the authors could not propose a demonstrator to experimentally validate their work. The authors should be inspired by the following work, with analysis and citation: https://doi.org/10.3390/en15031194

5) What are the limitations of the proposed approach? A discussion is imperative to step back from this topic.

6) The general conclusion must be completely reworked because it does not comply with the standards: reminder of the context and objectives of the study; quantitative synthesis of the main results obtained and adequacy with the research questions identified; work perspectives.

7) The figures are of poor quality. It is very difficult to read them.

Author Response

Dear Sir,

Greetings of the Day!

The authors express their sincere acknowledgements and thanks to the honourable editor and worthy reviewers for devoting their valuable time to review the manuscript and providing necessary comments and suggestions. These suggestions and comments have undoubtedly contributed to improvement in the overall quality of the paper. The authors have taken the comments very positively and have tried to answer each comment in the best possible way. The comments and suggestions of the honourable editor and reviewers have been incorporated in the revised manuscript.

Reviewer 5 Report

·      The paper describes that the use of ICE-based vehicles have been resulting into acute issues like air pollutions, global warming, and depletion of the world’s petroleum resources. ICEs are to be replaced by Fuel Cell Vehicles, Electric Vehicles, and Hybrid Electric Vehicles, saying further that the performance of HEV has been drawing more attention the others over the past few years.

·      Specific type of gasoline is used as a fuel in a conventional hybrid electric car, which saves fuel consumption in a variety of ways including collecting energy during braking, downsizing the engine more effectively, and turning engine off when it is not in use. Regenerative is used in HEV for converting the vehicle kinetic energy into electric energy to charge batteries  rather than letting it to dissipate as heat in standard brake; also in HEV are utilizing a component called  super capacitor with high energy density that are used to store energy and release regenerative energy. Typical basic block diagram of HEV is shown in Figure 1

·      The main contribution of the paper is the use of ANN is to control the Bidirectional DC-DC Converter in the two operation modes of powering from two voltage sources (FVS and SVS). The performance, as the authors claim, is compared of the ANN-based approach to that of the PID-control traditional approach.

·      Figure 2 shows the Dual-battery energy storage (VFVS, CFVS, VSVS, CSVS) in BDC architecture which when coupled with ANN control makes the whole theme of the paper

·      The BCD multiple working modes are shown in the flowchart of Figure 12 when the vehicle is moving, then it operates in either accelerating mode or braking mode, else it returns to reassign mode to operate the next prescience of switching mode. Voltages VFVS(96V) and VSVS(96V) are used perceive mode change in the halting condition

Concern 1.       If VFVS<96V is YES, then the control moves into Dual low-voltage sources act at Boost mode (SVS to FVS) mode

Concern 2.       In the case of NO, it is equal or more than 96V, then how the test of VSVS<48V becomes applicable for making dual voltage source into Buck mode (  because the voltage is larger than 96Vwhich is obviously not a possible condition

Concern 3.       The flowchart needs be made understandable with a references to Figure 1 and Figure 2

·      Output voltages and Inductor currents in in Regenerative mode are given in Figure 16 in regenerative mode while Figure 18 and Figure 22  shows Gate signals, output voltage and inductor currents for boost and buck mode respectively.

·      Regulated current step variations in inductor currents at high bus voltage results are compared between PID and ANN control in Figure 24 and results related to Regulated current step variations in inductor currents at boost mode at Boost and Buck mode are given respectively in Figure 25 and Figure 26 respectively

Concern 4.       The abstract needs to be structured, consisting of background and motivation, stating the problem and its context; technique and results, and clearly state the novelty/contribution of the paper and should include experimental results of the ANN-based control in your paper.

Concern 5.       The introduction is recommended to re-write as a four paragraphs piece: 1) State the background and motivation and describe the problem to solve. 2) Make a revision of the state-of-the-art with reference to published key works related in general to the work, then 3) more specific work to present your work in a genuinely presented context to other published methodologies, 4) Describe the technique, and 5) Clearly state the novelty/contribution of the proposed methodology and briefly describe how it is validated in context to other methodologies. It must be noticed that it is required to present results from actual experimental data.

Is the subject matter presented in a comprehensive manner?

·      The 19-page paper is based heavily on Figure 2 and the control of power flow and to genuinely support the title “A Smart ANN Based Converter for Efficient Bidirectional Power Flow in Hybrid Electric Vehicles”. The theory support and related explanation is comprehensively covered.

·      The power flow is described in the well-explained explanatory diagrams under dual low-voltage sources powering mode as much as is explained in the Regenerative mode at high voltage in support of the title.

·      The method is explained by giving block diagram of HEV in Figure 1 to support the title and the simulation results are provided to compare the traditional PID control with the ANN-based control.

Are the references provided applicable and sufficient?

·      The authors take support from twenty four (24) mostly recently and related references and including two from MDPI Energies. The results are related to turbine modelling and present the practical utility of the results for various industrial applications.

·      It is a good paper and support the title well with utility for use in HEV applications. I decide to accept the paper for publication in the MDPI Electronics.

Concern 6.       However, the results need to be compared with 1-2 benchmark papers in the reported literature to present the contribution of the paper more effectively.

Concern 7.       All diagrams be drawn with 300dpi line sizes, and text on the diagrams should be of the same font and size similar to the caption.

Author Response

(The authors gave the same response as above.)

Round 2

Reviewer 1 Report

The authors should further enlarge the Introduction with current work about advanced control and optimization algorithms.

Improve the quality of figures

Reviewer 2 Report

The revised manuscript is okay with me.

Author Response

We the authors would like to thank honorable reviewer for the suggestion made. English language spell check is performed and revisions are done. Thank you. 

Reviewer 3 Report

the authors have fully responded to my suggestions. The paper can now be published

Author Response

We the authors would like to thank honorable reviewer for the suggestions made. all the necessary modifications are made in the revised manuscript. Thank you.

Reviewer 4 Report

The authors have responded to the majority of my comments and I congratulate them.

However, there are a few minor points to correct.

Please rework the following figures as the annotations and/or axis titles of the graphs are unreadable: Figure 1, Figure 2, Figure 3, Figure 5, Figure 11, and Figure 12. For example, you can use PowerPoint to rework the text of a figure. Once the figure is enhanced, you can save it as an image.

Author Response

We the authors would like to thank honorable reviewer for the suggestions             made. Figure 1, Figure 2, Figure 3, Figure 5, Figure 11, and Figure 12 are edited and the results are clearly presented.  English language and style are verified and necessary modifications are made. Thank You.